# EARLY STOPPING BY GRADIENT DISPARITY

## ABSTRACT

Validation-based early-stopping methods are one of the most popular techniques used to avoid over-training deep neural networks. They require to set aside a reliable unbiased validation set, which can be expensive in applications offering limited amounts of data. In this paper, we propose to use *gradient disparity*, which we define as the $\ell_2$ norm distance between the gradient vectors of two batches drawn from the training set. It comes from a probabilistic upper bound on the difference between the classification errors over a given batch, when the network is trained on this batch and when the network is trained on another batch of points sampled from the same dataset. We empirically show that gradient disparity is a very promising early-stopping criterion when data is limited, because it uses all the training samples during training. Furthermore, we show in a wide range of experimental settings that gradient disparity is not only strongly related to the usual generalization error between the training and test sets, but that it is also much more informative about the level of label noise.

## 1    INTRODUCTION

Early stopping is a commonly used regularization technique to avoid under/over fitting deep neural networks trained with iterative methods, such as gradient descent (Prechelt, 1998; Yao et al., 2007; Gu et al., 2018). To have an unbiased proxy on the generalization error, early stopping requires a separate accurately labeled validation set. However, labeled data collection is an expensive and time consuming process that might require domain expertise (Roh et al., 2019). Moreover, deep learning is becoming popular to use for new and critical applications for which there is simply not enough available data. Hence, it is advantageous to have a signal of overfitting that does not require a validation set, then all the available data can be used for training the model.

Let $S_1$ and $S_2$ be two batches of points sampled from the available (training) dataset. Suppose that $S_1$ is selected for an iteration (step) of stochastic gradient descent (SGD), which then updates the parameter vector to $w_1$. The average loss over $S_1$ is in principle reduced, given a sufficiently small learning rate. However, the average loss over the other batch $S_2$ (i.e., $L_{S_2}(h_{w_1})$) is not as likely to be reduced. It will remain on average larger than the loss computed over $S_2$, if it was $S_2$ instead of $S_1$ that had been selected for this iteration (i.e., $L_{S_2}(h_{w_2})$). The difference is the penalty $\mathcal{R}_2$ that we pay for choosing $S_1$ over $S_2$ (and similarly, $\mathcal{R}_1$ is the penalty that we would pay for choosing $S_2$ over $S_1$). $\mathcal{R}_2$ is illustrated in Figure 1 for a hypothetical non-convex loss as a function of a one dimensional parameter. The expected penalty measures how much, in an iteration, a model updated on one batch ($S_1$) is able to generalize on average to another batch ($S_2$) from the dataset. Hence, we call $\mathcal{R}$ the generalization penalty.

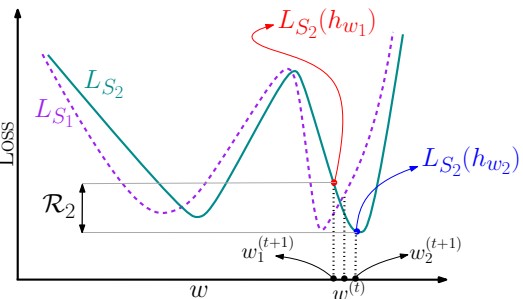

Figure 1: An illustration of the penalty term $\mathcal{R}_2$, where the y-axis is the loss, and the x-axis indicates the parameters of the model. $L_{S_1}$ and $L_{S_2}$ are the average losses over batches $S_1$ and $S_2$, respectively. $w^{(t)}$ is the parameter at iteration $t$ and $w_i^{(t+1)}$ is the parameter at iteration $t+1$ if batch $S_i$ was selected for the update step at iteration $t$, with $i \in \{1, 2\}$.

We establish a probabilistic upper-bound on the sum of the expected penalties $\mathbb{E}[\mathcal{R}_1] + \mathbb{E}[\mathcal{R}_2]$ by adapting the PAC-Bayesian framework (McAllester, 1999a;b; 2003) given a pair of batches $S_1$ and $S_2$ sampled from the dataset (Theorem 1). Interestingly, under some mild assumptions, this upper bound is essentially a simple expression driven by $\|g_1 - g_2\|_2$, where $g_1$ and $g_2$ are the gradient vectors over the two batches $S_1$ and $S_2$, respectively. We call this *gradient disparity*: it measures how a small gradient step on one batch negatively affects the performance on another one.

Gradient disparity is simple to use and it is computationally tractable during the course of training. Our experiments on state-of-the-art configurations suggest a very strong link between gradient disparity and generalization error; we propose gradient disparity as an effective early stopping criterion. Gradient disparity is particularly useful when the available dataset has limited labeled data, because it does not require splitting the available dataset into training and validation sets so that all the available data can be used during training, unlike for instance $k$-fold cross validation. We observe that using gradient disparity, instead of an unbiased validation set, results in at least $1\%$ predictive performance improvement for critical applications with limited and very costly available data, such as the MRNet dataset that is a small size image-classification dataset used for detecting knee injuries (Table 1).

| Task | Method | Test loss | Test AUC score (in percentage) |
|---|---|---|---|
| abnormal | 5-fold CV | $0.284 \pm 0.016$ ($0.307 \pm 0.057$) | $71.016 \pm 3.66$ ($87.44 \pm 1.35$) |
| | GD | $\mathbf{0.274} \pm 0.004$ ($\mathbf{0.275} \pm 0.053$) | $\mathbf{72.67} \pm 3.85$ ($\mathbf{88.12} \pm 0.35$) |
| ACL | 5-fold CV | $0.973 \pm 0.111$ ($1.246 \pm 0.142$) | $79.80 \pm 1.23$ ($89.32 \pm 1.47$) |
| | GD | $\mathbf{0.842} \pm 0.101$ ($\mathbf{1.136} \pm 0.121$) | $\mathbf{81.81} \pm 1.64$ ($\mathbf{91.52} \pm 0.09$) |
| meniscal | 5-fold CV | $0.758 \pm 0.04$ ($1.163 \pm 0.127$) | $73.53 \pm 1.30$ ($72.14 \pm 0.74$) |
| | GD | $\mathbf{0.726} \pm 0.019$ ($\mathbf{1.14} \pm 0.323$) | $\mathbf{74.08} \pm 0.79$ ($\mathbf{73.80} \pm 0.24$) |

Table 1: The loss and area under the receiver operating characteristic curve (AUC score) on the MRNet test set (Bien et al., 2018), comparing 5-fold cross validation (5-fold CV) and gradient disparity (GD), when both are used as early stopping criteria for detecting the presence of abnormally, ACL tears, and meniscal tears from the sagittal plane MRI scans. The corresponding curves during training are shown in Figure 10. The results of early stopping are given, both when the metric has increased for 5 epochs from the beginning of training and between parenthesis when the metric has increased for 5 consecutive epochs.

Moreover, when the available dataset contains noisy labels, the validation set is no longer a reliable predictor of the clean test set (see e.g., Figure 9 (a) (left)), whereas gradient disparity correctly predicts the performance on the test set and again can be used as a promising early-stopping criterion. Furthermore, we observe that gradient disparity is a better indicator of label noise level than generalization error, especially at early stages of training. Similarly to the generalization error, it decreases with the training set size, and it increases with the batch size.

**Paper Outline.** In Section 2, we formally define the generalization penalty. In Section 3, we give the upper bound on the generalization penalty. In Section 4, we introduce the gradient disparity metric. In Section 5, we present experiments that support gradient disparity as an early stopping criterion. In Section 6, we assess gradient disparity as a generalization metric. Finally, in Section 7, we further discuss the observations and compare gradient disparity to related work. A detailed comparison to related work is deferred to Appendix H. For our experiments, we consider four image classification datasets: MNIST, CIFAR-10, CIFAR-100 and MRNet, and we consider a wide range of neural network architectures: ResNet, VGG, AlexNet and fully connected neural networks.

## 2 GENERALIZATION PENALTY

Consider a classification task with input $x \in \mathcal{X} := \mathbb{R}^n$ and ground truth label $y \in \{1, 2, \cdots, k\}$, where $k$ is the number of classes. Let $h_w \in \mathcal{H} : \mathcal{X} \to \mathcal{Y} := \mathbb{R}^k$ be a predictor (classifier) parameterized by the parameter vector $w \in \mathbb{R}^d$, and $l(\cdot, \cdot)$ be the 0-1 loss function $l(h_w(x), y) = \mathbb{1}[h_w(x)[y] < \max_{j \neq y} h_w(x)[j]]$ for all $h_w \in \mathcal{H}$ and $(x, y) \in \mathcal{X} \times \{1, 2, \cdots, k\}$. The expected loss and the empirical loss over the training set $S$ of size $m$ are respectively defined as

$$L(h_w) = \mathbb{E}_{(x,y)\sim D}\left[l\left(h_w(x), y\right)\right] \qquad \text{and} \qquad L_S(h_w) = \frac{1}{m}\sum_{i=1}^{m} l(h_w(x_i), y_i), \qquad (1)$$

where $D$ is the probability distribution of the data points and $(x_i, y_i)$ are i.i.d. samples drawn from $S \sim D^m$. $L_S(h_w)$ is also called the training classification error. Similar to the notation used in (Dziugaite & Roy, 2017), distributions on the hypotheses space $\mathcal{H}$ are simply distributions on the underlying parameterization. With some abuse of notation, $\nabla L_{S_i}$ refers to the gradient with respect to the surrogate differentiable loss function, which in our experiments is the cross entropy.

In a mini-batch gradient descent (SGD) setting, consider two batches of points, denoted by $S_1$ and $S_2$, which have respectively $m_1$ and $m_2$ number of samples, with $m_1 + m_2 \leq m$. The average loss functions over these two sets of samples are $L_{S_1}(h_w)$ and $L_{S_2}(h_w)$, respectively. Let $w = w^{(t)}$ be the parameter vector at the beginning of an iteration $t$. If $S_1$ is selected for the next iteration, $w$ gets updated to $w_1 = w^{(t+1)}$ with

$$w_1 = w - \gamma \nabla L_{S_1}(h_w), \qquad (2)$$

where $\gamma$ is the learning rate. Conversely, if $S_2$ had been selected instead of $S_1$, the updated parameter vector at the end of this iteration would have been $w_2 = w - \gamma \nabla L_{S_2}(h_w)$. Therefore, the generalization penalty on batch $S_2$ is defined as $\mathcal{R}_2 = L_{S_2}(h_{w_1}) - L_{S_2}(h_{w_2})$, which is the gap between the loss over $S_2$, $L_{S_2}(h_{w_1})$, and its target value, $L_{S_2}(h_{w_2})$, at the end of iteration $t$.

When selecting $S_1$ for the parameter update, Equation (2) makes a step towards learning the input-output relations of batch $S_1$. If this negatively affects the performance on batch $S_2$, $\mathcal{R}_2$ will be large; the model is learning the data structures that are unique to $S_1$ and that do not appear in $S_2$. Because $S_1$ and $S_2$ are batches of points sampled from the same distribution $D$, they have data structures in common. If, throughout the learning process, we consistently observe that, in each update step, the model learns structures unique to only one batch, then it is very likely that the model is memorizing the labels instead of learning the common data-structures. This is captured by the generalization penalty $\mathcal{R}$.

## 3 BOUND ON THE GENERALIZATION PENALTY

We adapt the PAC-Bayesian framework (McAllester, 1999a;b) to account for the trajectory of the learning algorithm; For each learning iteration we define a prior, and two possible posteriors depending on the choice of the batch selection. Let $w \sim P$ be an initial parameter vector that follows a prior distribution $P$ which is a $\mathcal{F}_t$-measurable function, where $\mathcal{F}_t$ denotes the filtration of the available information at the beginning of iteration $t$. Let $h_{w_1}, h_{w_2}$ be the two learned single predictors, at the end of iteration $t$, from $S_1$ and $S_2$, respectively. In this framework, for $i \in \{1, 2\}$, each predictor $h_{w_i}$ is randomized and becomes $h_{\nu_i}$ with $\nu_i = w_i + u_i$, where $u_i$ is a random variable whose distribution might depend on $S_i$. Let $Q_i$ be the distribution of $\nu_i$, which is a distribution over the predictor space $\mathcal{H}$ that depends on $S_i$ via $w_i$ and possibly $u_i$. Let $\mathcal{G}_i$ be a $\sigma$-field such that $\sigma(S_i) \cup \mathcal{F}_t \subset \mathcal{G}_i$ and that the posterior distribution $Q_i$ is $\mathcal{G}_i$-measurable for $i \in \{1, 2\}$. We further assume that the random variable $\nu_1 \sim Q_1$ is statistically independent from the draw of the batch $S_2$ and, vice versa, that $\nu_2 \sim Q_2$ is independent from the batch $S_1$[1], i.e., $\mathcal{G}_1 \perp\!\!\!\perp \sigma(S_2)$ and $\mathcal{G}_2 \perp\!\!\!\perp \sigma(S_1)$.

**Theorem 1.** *For any $\delta \in (0, 1]$, with probability at least $1 - \delta$ over the sampling of sets $S_1$ and $S_2$, the sum of the expected penalties conditional on $S_1$ and $S_2$, respectively, satisfies*

$$\mathbb{E}\left[\mathcal{R}_1\right] + \mathbb{E}\left[\mathcal{R}_2\right] \leq \sqrt{\frac{2KL(Q_2||Q_1) + 2\ln\frac{2m_2}{\delta}}{m_2 - 2}} + \sqrt{\frac{2KL(Q_1||Q_2) + 2\ln\frac{2m_1}{\delta}}{m_1 - 2}}. \qquad (3)$$

Theorem 1, whose proof is given in Appendix B, shows why generalization penalties are better suited to our setting where the two batches $S_1$ and $S_2$ are both drawn from the training set $S$ than the usual generalization errors. After an iteration, the network learns a posterior distribution $Q_1$ on its parameters from $S_1$, yielding to the parameter vector $\nu_1 \sim Q_1$. The expected generalization

---

[1] Batches $S_1$ and $S_2$ are drawn without replacement, and the random selection of indices of batches $S_1$ and $S_2$ is independent from the dataset $S$. Hence, similarly to Negrea et al. (2019); Dziugaite et al. (2020), we have $\sigma(S_1) \perp\!\!\!\perp \sigma(S_2)$.

error at that time is defined as $\text{GE}_1 = \mathbb{E}_{\nu_1 \sim Q_1}[L(h_{\nu_1})] - \mathbb{E}_{\nu_1 \sim Q_1}[L_{S_1}(h_{\nu_1})]$. In practice, $L(h_{\nu_1})$ is estimated by the test loss over a batch of unseen data, which is independent from $\nu_1 \sim Q_1$. If $S_2$ is this batch, then[2] $\text{GE}_1 \approx \mathbb{E}_{\nu_1 \sim Q_1}[L_{S_2}(h_{\nu_1})] - \mathbb{E}_{\nu_1 \sim Q_1}[L_{S_1}(h_{\nu_1})]$. However, this estimate requires to set $S_2$ aside from $S$ not only during that step but also during all the previous steps, because otherwise the model $h_{\nu_1}$ would not be independent from $S_2$, making the estimate of $\text{GE}_1$ biased. Therefore $S_2$ must be sampled from the validation set, and cannot be used during training. In contrast, Theorem 1 is valid even if the trained model $h_{\nu_1}$ depends on the samples within the batch $S_2$. Therefore, the bound on the sum of the (expected) generalization penalties does no longer require to set $S_2$ aside from $S$ in previous iterations; all data previously reserved for validation can now be used for training. This is what makes these penalties appealing to measure generalization especially when the available dataset is limited and/or noisy, as we will see in Section 5.

Theorem 1 remains valid if batch $S_2$ was sampled from the validation set, in which case it can be compared with known generalization error (GE) bounds, as now $h_{\nu_1}$ does not depend on samples of $S_2$. Similarly to $\text{GE}_1$, let $\text{GE}_2$ be the generalization error when $S_2$ is the training set, while $S_1$ is the test set. By adding $\text{GE}_1$ and $\text{GE}_2$ we obtain $\mathbb{E}[\mathcal{R}_1] + \mathbb{E}[\mathcal{R}_2]$, hence Theorem 1 also upper bounds an estimate of $\text{GE}_1 + \text{GE}_2$. We could have obtained another upper bound by directly applying the bounds from (McAllester, 2003; Neyshabur et al., 2017b), which gives

$$\mathbb{E}[\mathcal{R}_1] + \mathbb{E}[\mathcal{R}_2] \leq 2\sqrt{\frac{2\text{KL}(Q_2||P) + 2\ln\frac{2m_2}{\delta}}{m_2 - 1}} + 2\sqrt{\frac{2\text{KL}(Q_1||P) + 2\ln\frac{2m_1}{\delta}}{m_1 - 1}}. \tag{4}$$

The main difference between Equations (3) and (4) is that the former needs the difference only between the two posterior distributions, $Q_1$ and $Q_2$, whereas the latter requires the difference between the prior distribution $P$ and the posterior distributions $Q_1$ and $Q_2$. Besides the minor difference of the multiplicative factor 2, the upper bound in Equation (3) is non-vacuous for a larger class of distributions than the upper bound in Equation (4): When $Q_1$ and $Q_2$ are close to each other but not to $P$, the upper bound in Equation (3) is much tighter than the one in Equation (4). Moreover, in the next section, we show that, under reasonable assumptions, the upper bound in Equation (3) boils down to a very tractable generalization metric that we call gradient disparity.

## 4  GRADIENT DISPARITY

The randomness modeled by $u_i$, conditioned on the current batch $S_i$, comes from (i) the parameter vector at the beginning of the iteration $w$, which itself comes from the random parameter initialization and the stochasticity of the parameter updates until that iteration, and (ii) the gradient vector $\nabla L_{S_i}$, which may also be random because of the possible additional randomness in the network structure due for instance to dropout (Srivastava et al., 2014). A common assumption made in the literature is that the random perturbation $u_i$ follows a normal distribution (Bellido & Fiesler, 1993; Neyshabur et al., 2017b). The upper bound in Theorem 1 takes a particularly simple form if we assume that for $i \in \{1, 2\}$ the random perturbations $u_i$ are zero mean i.i.d. normal distributions ($u_i \sim \mathcal{N}(0, \sigma^2 I)$), and that $w_i$ is fixed, as in the setting of (Dziugaite & Roy, 2017). $\text{KL}(Q_1||Q_2)$ is then the KL-divergence between two multivariate normal distributions .

Let us denote $\nabla L_{S_1}(h_w)$ by $g_1 \in \mathbb{R}^d$ and $\nabla L_{S_2}(h_w)$ by $g_2 \in \mathbb{R}^d$. As $w_i = w - \gamma g_i$ for $i \in \{1, 2\}$, the KL-divergence between $Q_1 = \mathcal{N}(w_1, \sigma^2 I)$ and $Q_2 = \mathcal{N}(w_2, \sigma^2 I)$ (Lemma 1 in Appendix A) is simply

$$\text{KL}(Q_1||Q_2) = \frac{1}{2}\frac{\gamma^2}{\sigma^2}\|g_1 - g_2\|_2^2 = \text{KL}(Q_2||Q_1), \tag{5}$$

which shows that, keeping a constant step size $\gamma$ and assuming the same variance for the random perturbations $\sigma^2$ in all the steps of the training, the bound in Theorem 1 is driven by $\|g_1 - g_2\|_2$.

---

[2]More formally,

$$\text{GE}_1 = \mathbb{E}_{\nu_1 \sim Q_1}[L_{S_2}(h_{\nu_1})] - \mathbb{E}_{\nu_1 \sim Q_1}[L_{S_1}(h_{\nu_1})] + \overbrace{(\mathbb{E}_{\nu_1 \sim Q_1}[L(h_{\nu_1})] - \mathbb{E}_{\nu_1 \sim Q_1}[L_{S_2}(h_{\nu_1})])}^{\zeta}$$

where from Hoeffding's bound (Theorem 2 in Appendix A) $\mathbb{P}(|\zeta| \geq t) \leq \exp(-2m_2 t^2)$, and $\text{GE}_1$ is approximated with the first term.

This indicates that the smaller the $\ell_2$ distance between gradient vectors is, the lower the upper bound on the generalization penalty is, and therefore the closer the performance of a model trained on one batch is to a model trained on another batch.

For two batches of points $S_i$ and $S_j$, with gradient vectors $g_i$ and $g_j$, respectively, we define the *gradient disparity* (GD) between $S_i$ and $S_j$ as

$$\mathcal{D}_{i,j} = \|g_i - g_j\|_2 . \tag{6}$$

Gradient disparity is empirically tractable, and provides a probabilistic guarantee on the sum of the generalization penalties of $S_i$ and $S_j$, modulo the Gaussianity assumptions made in this section. Gradient disparity can be computed within batches of the training or the validation set. As discussed in Section 3, we focus on the first case to have a generalization metric that does not require validation data and that provides an early stopping criterion with all the available data used for training.

We focus on the vanilla SGD optimizer. In Appendix G, we extend the analysis to other adaptive optimizers: SGD with momentum (Qian, 1999), Adagrad (Duchi et al., 2011), Adadelta (Zeiler, 2012), and Adam (Kingma & Ba, 2014). In all these optimizers, we observe that gradient disparity (Equation (6)) appears in $\mathrm{KL}(Q_1 \| Q_2)$ with other factors that depend on a decaying average of past gradient vectors. Experimental results support the use of gradient disparity as an early stopping metric also for these popular optimizers (see Figure 22 in Appendix G).

Computing the gradient disparity averaged over $B$ batches requires all the $B$ gradient vectors at each iteration, which is computationally expensive if $B$ is large. We approximate it by computing it over only a much smaller subset of the batches, of size $s \ll B$, $\overline{\mathcal{D}} = \sum_{i=1}^{s} \sum_{j=1, j \neq i}^{s} \mathcal{D}_{i,j} / s(s-1)$. In the experiments presented in this paper, $s = 5$; we observed that such a small subset is already sufficient (see Appendix C.2 for an experimental comparison of different values of $s$). We also present an alternative approach that relies on the distribution of gradient disparity instead of its average in Appendix F to have a finer-grain signal of overfitting.

As training progresses, the gradient magnitude starts to decrease, and therefore the value of gradient disparity might decrease not necessarily because the distance between two gradient vectors is decreasing, but because their magnitudes are decreasing. Hence, in order to compare different stages of training, we re-scale the loss values within each batch, before computing gradient disparity in $\overline{\mathcal{D}}$ (refer to Appendix C.1 for more details). Moreover, we consider the mean square error (MSE) for the choice of the surrogate loss in Appendix C.3 and we observe that gradient disparity is positively correlated with the MSE test loss as well.

## 5    Gradient Disparity as an Early Stopping Criterion

**Comparison to $k$-fold Cross Validation.** Early stopping is a popular technique used in practice to avoid overfitting (Prechelt, 1998; Yao et al., 2007; Gu et al., 2018). The optimization is stopped when the performance of the model on a validation set starts to diverge from its performance on the training set. Early stopping is of particular interest in the presence of label noise, because the model first learns the samples with correct labels, and next the corrupted samples (Li et al., 2019). To emphasize the particular application of gradient disparity, we compare it to $k$-fold cross validation in two settings: (i) when the available dataset is limited and (ii) when the available dataset has corrupted labels. We simulate the limited data scenario by using a small subset of three image classification benchmark datasets: MNIST, CIFAR-10 and CIFAR-100, and the noisy labeled data scenario is simulated by using a corrupted version of these datasets. We also evaluate gradient disparity on a medical dataset (MRNet dataset) with limited available data and in this setting, we use the entire dataset for training.

(i) We compare gradient disparity with $k$-fold cross validation (Stone, 1974) used as an early stopping criterion in Table 2 (top) when there is limited labeled data. We observe that gradient disparity performs well as an early stopping criterion, especially when the data is complex (CIFAR-100 is more complex than CIFAR-10). As it uses every available sample, instead of only a $1 - 1/k$ portion of the dataset, it results in a better performance on the final unseen (test) data (see also Table 4 and Figure 8 in Appendix D).

In many real-world applications, collecting (labeled) data is very costly. In some medical applications, it requires high costs of patient data collection and medical staff expertise. For an example

of such an application, we consider the MRNet dataset (Bien et al., 2018), which contains limited number of MRI scans to study the presence of abnormally, ACL tears and meniscal tears in knee injuries. We observe that using gradient disparity instead of a validation set, results in over $1\%$ improvement (on average over all three tasks) in the test AUC score, and therefore additional correct detection for more than one patient for each task (see Table 1 and Figure 10 in Appendix D).

(ii) When the labels of the available data are noisy, the validation set is no longer a reliable estimate of the unseen set (this can be clearly observed in Figure 9 (left column)). Nevertheless, and although it is computed over the noisy training set, gradient disparity reflects the performance on the test set quite well (Figure 9 (middle left column)). As a result (see Table 2 (bottom)), gradient disparity performs better than $k$-fold cross validation as an early stopping criterion. The same applies to two other datasets (see Table 6 and Figure 9 in Appendix D).

| Label noise | Method | Test loss | Test accuracy | Top-5 test accuracy |
|---|---|---|---|---|
| 0% | 5-fold CV | $4.249 \pm 0.028$ | $6.79 \pm 0.49$ | $22.19 \pm 0.77$ |
| | GD | $\mathbf{4.057} \pm 0.043$ | $\mathbf{9.99} \pm 0.92$ | $\mathbf{27.84} \pm 1.30$ |
| 50% | 10-fold CV | $5.023 \pm 0.083$ | $1.59 \pm 0.15$ | $6.47 \pm 0.52$ |
| | GD | $\mathbf{4.463} \pm 0.038$ | $\mathbf{3.68} \pm 0.52$ | $\mathbf{15.22} \pm 1.24$ |

Table 2: The final test loss and accuracy when using gradient disparity (GD) and $k$-fold cross validation (CV) as early stopping criteria, (top) when the available dataset is limited and (bottom) when the available data has noisy labeled samples. To simulate a limited-data scenario, we consider as a training set a subset of 1280 samples of the CIFAR-100 dataset. The configurations in the top row and the bottom row are ResNet-34 and ResNet-18, respectively. In both methods, the optimization is stopped when the metric (validation loss or GD) increases for 5 epochs.

## 6 VALIDATING GRADIENT DISPARITY AS A GENERALIZATION METRIC

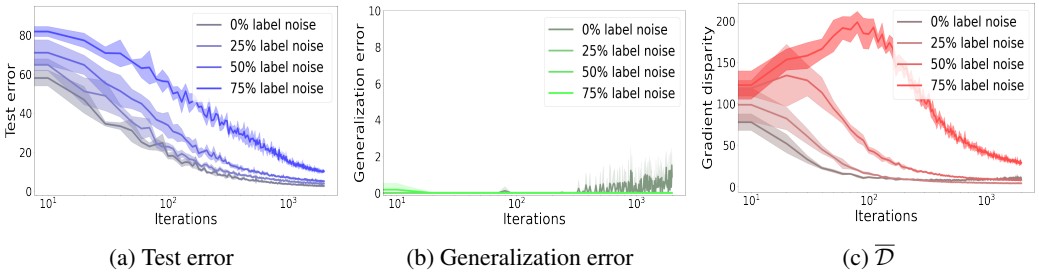

(a) Test error   (b) Generalization error   (c) $\overline{\mathcal{D}}$

Figure 2: The error percentage and $\overline{\mathcal{D}}$ during training with different amounts of randomness in the training labels for an AlexNet trained on a subset of 12.8 k points of the MNIST training dataset. Pearson's correlation coefficient between gradient disparity and test error (TE)/test loss (TL) over all the iterations and over all levels of randomness are $\rho_{\overline{\mathcal{D}},\text{TE}} = 0.861$ and $\rho_{\overline{\mathcal{D}},\text{TL}} = 0.802$. The generalization error (gap) is the difference between the train and test errors.

In this section, we demonstrate that factors that contribute to improve or degrade the generalization performance of a model (e.g., label noise level, training set size and batch size), have an often strikingly similar effect on the value of gradient disparity as well.

**Label Noise Level.** Deep neural networks, trained with the SGD algorithm, achieve excellent generalization performance (Hardt et al., 2015), while achieving zero training error on randomly labeled data in classification tasks (Zhang et al., 2016). Understanding what distinguishes the model when it is trained on correct labels and when it is trained on randomly labeled data is still an evolving area of research. We conjecture that as label noise level increases, the gradient vectors diverge more. Hence, when the network is trained with correctly labeled samples, the gradient disparity is low, whereas when it is trained with corrupted samples the gradient disparity is high. The experimental results support this conjecture in a wide range of settings and show that gradient disparity is indeed very sensitive to label noise level (see also Figures 12, 15, 19 and 20).

Figure 2 shows the test error for networks trained with different amounts of label noise. Interestingly, observe that for this setting the test error for the network trained with $75\%$ label noise remains relatively small, indicating the good resistance of the model against memorization of corrupted samples. As suggested both from the test error (Figure 2 (a)) and the average gradient disparity (Figure 2 (c)), there is no proper early stopping time for these experiments. The generalization error (Figure 2 (b)) remains close to zero, regardless of the level of label noise, and hence fails to account for label noise. In contrast, the average gradient disparity is very sensitive to the label noise level in all stages of training as shown in Figure 2 (c), as desired for a metric measuring generalization.

**Training Set Size.** The test error decreases with the size of the training set (Figure 3 (left)) and a reliable generalization metric should therefore reflect this property. Many of the previous metrics fail to do so, as shown by (Neyshabur et al., 2017a; Nagarajan & Kolter, 2019). In contrast, the average gradient disparity indeed clearly decreases with the size of the training set, as shown in Figure 3 (right) (see also Figure 17 in Appendix E).

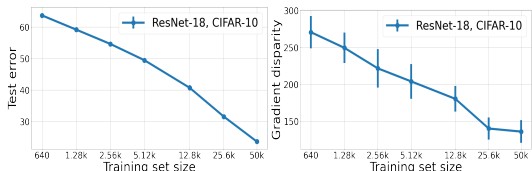

Figure 3: The test error (TE) and average gradient disparity ($\overline{\mathcal{D}}$) for networks that are trained (until reaching the training loss value of $0.01$) over training sets with different sizes. We observe a very strong positive correlation: $\rho_{\overline{\mathcal{D}}, \text{ TE}} = 0.984$.

**Batch Size.** In practice, the test error increases with the batch size (Figure 4 (left)). We observe that gradient disparity also increases with the batch size (Figure 4 (right)). This observation is counter-intuitive because one might expect that gradient vectors get more similar when they are averaged over a larger batch. This might be the explanation behind the decrease in gradient disparity from batch size 256 to 512 for the VGG-19 network. Observe also that gradient disparity correctly predicts that VGG-19 generalizes better than ResNet-34 for this dataset. Gradient disparity matches the ranking of test errors for different networks, trained with different batch sizes, as long as the batch sizes are not too large (see also Figure 18 in Appendix E).

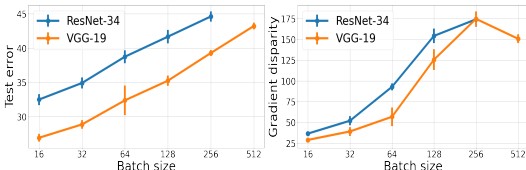

Figure 4: The test error and average gradient disparity for networks that are trained with different batch sizes. A ResNet-34 and a VGG-19 network that are trained on the CIFAR-10 dataset. The correlation between $\overline{\mathcal{D}}$ and test error (TE) for ResNet-34, VGG-19, and both graphs combined are $\rho_{\overline{\mathcal{D}}, \text{ TE}} = 0.985$, $\rho_{\overline{\mathcal{D}}, \text{ TE}} = 0.926$, and $\rho_{\overline{\mathcal{D}}, \text{ TE}} = 0.893$, respectively.

**Width.** In practice, the test error has been observed to decrease with the network width. We observe that gradient disparity (normalized with respect to the number of parameters) also decreases with network width for ResNet, VGG and fully connected neural networks that are trained on the CIFAR-10 dataset (see Figure 14 in Appendix E.2).

## 7 DISCUSSION AND RELATED WORK

Finding a practical metric that completely captures the generalization properties of deep neural networks, and in particular indicates the level of randomness in the labels and decreases with the size of the training set, is still an active research direction (Dziugaite & Roy, 2017; Neyshabur et al., 2017a; Nagarajan & Kolter, 2019). A very recent line of work assesses the similarity between the gradient updates of two batches (samples) in the training set. The coherent gradient hypothesis (Chatterjee, 2020) states that the gradient is stronger in directions where similar examples exist and towards which the parameter update is biased. He & Su (2020) presents the local elasticity phenomenon, which measures how prediction over one sample changes, as the network is updated on another sample. The generalization penalty introduced in our work measures how the prediction over one sample (batch) changes when the network is updated on the same sample instead of being updated on another sample, which can signal overfitting implicitly within the training set.

Tracking generalization by measuring the similarity between gradient vectors is particularly beneficial as it is empirically tractable during training and does not require access to unseen data. Sankararaman et al. (2019) proposes gradient confusion, which is a bound on the inner product

of two gradient vectors, and shows that the larger the gradient confusion is, the slower the convergence takes place. Gradient interference (when the inner product of gradient vectors is negative) has been studied in multi-task learning, reinforcement learning and temporal difference learning (Riemer et al., 2018; Liu et al., 2019; Bengio et al., 2020). Yin et al. (2017) studies the relation between gradient diversity, which measures the dissimilarity between gradient vectors, and the convergence performance of distributed SGD algorithms. Fort et al. (2019) proposes a metric called stiffness, which is the cosine similarity between two gradient vectors, and shows empirically that it is related to generalization. Fu et al. (2020) studies the cosine similarity between two gradient vectors for natural language processing tasks. Mehta et al. (2020) measures the alignment between the gradient vectors within the same class (denoted by $\Omega_c$), and studies the relation between $\Omega_c$ and generalization as the scale of initialization is increased.

Another interesting line of work is the study of the variance of gradients in deep learning settings. Negrea et al. (2019) derives mutual information generalization error bounds for stochastic gradient Langevin dynamics (SGLD) as a function of the sum (over the iterations) of square gradient incoherences, which is closely related to gradients variance. Two-sample gradient incoherences also appear in Haghifam et al. (2020), there are taken between a training sample and a "ghost" sample that is not used during training and therefore taken from a validation set (unlike gradient disparity). The upper bounds in Negrea et al. (2019); Haghifam et al. (2020) are not intended to be used as early stopping criteria and are cumulative bounds that increase with the number of iterations. As shown in Appendix G, gradient disparity can be used as an early stopping criterion not only for SGD with additive noise (such as SGLD), but also other adaptive optimizers. Jastrzebski et al. (2020) studies the effect of the learning rate on the variance of gradients and hypothesizes that gradient variance counter-intuitively increases with the batch size, which is consistent with our observations. However, Qian & Klabjan (2020) shows that the variance of gradients is a decreasing function of the batch size. Jastrzebski et al. (2020); Qian & Klabjan (2020) mention the connection between variance of gradients and generalization as promising future directions. Our study shows that variance of gradients used as an early stopping criterion outperforms $k$-fold cross validation (see Table 8).

Mahsereci et al. (2017) proposes an early stopping criterion called evidence-based criterion (EB) that eliminates the need for a held-out validation set, similarly to gradient disparity. The EB-criterion is negatively related to the signal-to-noise ratio (SNR) of the gradient vectors. Liu et al. (2020) also proposes a relation between gradient SNR (called GSNR) and the one-step generalization error, with the assumption that both the training and the test sets are large, whereas gradient disparity targets limited datasets. Nevertheless, we have compared gradient disparity to these metrics (namely, EB, GSNR, gradient inner product, sign of the gradient inner product, variance of gradients, cosine similarity, and $\Omega_c$) in Appendix H. In Table 8, we observe that gradient disparity and variance of gradients used as early stopping criteria are the only metrics that consistently outperform $k$-fold cross validation, and that are more informative of the level of label noise compared to other metrics. We observe that the correlation between gradient disparity and the test loss is however in general larger than the correlation between variance of gradients and the test loss (Table 9).

A common drawback of the metrics based on the similarity between two gradient vectors, including gradient disparity, is that they are not informative when the gradient vectors are very small. In practice however, we observe (see for instance Figure 13) that the time at which the test and training losses start to diverge, which is the time when overfitting kicks in, does not only coincide with the time at which gradient disparity increases, but also occurs much before the training loss becomes infinitesimal. Hence, this drawback is unlikely to cause a problem for gradient disparity when it is used as an early stopping criterion. Nevertheless, Theorem 1 trivially holds when the gradient values are infinitesimal.

**Conclusion.** In this work, we propose gradient disparity, which is the $\ell_2$ norm of the difference between the gradient vectors of pairs of batches in the training set. Our empirical results on state-of-the-art configurations show indeed a strong link between gradient disparity and generalization error. Gradient disparity, similar to the test error, increases with the label noise level, decreases with the size of the training set and increases with the batch size. We therefore suggest, particularly when the available dataset is limited or noisy, gradient disparity as a promising early stopping criterion that does not require access to a validation set.

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

## A  ADDITIONAL THEOREM

Hoeffding's bound is used in the proof of Theorem 1, and Lemma 1 is used in Section 4.

**Theorem 2** (Hoeffding's Bound). *Let $Z_1, \cdots, Z_n$ be independent bounded random variables on $[a, b]$ (i.e., $Z_i \in [a, b]$ for all $1 \leq i \leq n$ with $-\infty < a \leq b < \infty$). Then*

$$\mathbb{P}\left(\frac{1}{n}\sum_{i=1}^{n}(Z_i - \mathbb{E}[Z_i]) \geq t\right) \leq \exp\left(-\frac{2nt^2}{(b-a)^2}\right)$$

*and*

$$\mathbb{P}\left(\frac{1}{n}\sum_{i=1}^{n}(Z_i - \mathbb{E}[Z_i]) \leq -t\right) \leq \exp\left(-\frac{2nt^2}{(b-a)^2}\right)$$

*for all $t \geq 0$.*

**Lemma 1** If $N_1 = \mathcal{N}(\mu_1, \Sigma_1)$ and $N_2 = \mathcal{N}(\mu_2, \Sigma_2)$ are two multivariate normal distributions in $\mathbb{R}^d$, where $\Sigma_1$ and $\Sigma_2$ are positive definite,

$$\text{KL}(N_1||N_2) = \frac{1}{2}\left(\text{tr}\left(\Sigma_2^{-1}\Sigma_1\right) - d + (\mu_2 - \mu_1)^T\Sigma_2^{-1}(\mu_2 - \mu_1) + \ln\left(\frac{\det\Sigma_2}{\det\Sigma_1}\right)\right).$$

## B  PROOF OF THEOREM 1

*Proof.* We compute the upper bound in Equation (3) using a similar approach as in McAllester (2003). The main challenge in the proof is the definition of a function $X_{S_2}$ of the variables and parameters of the problem, which can then be bounded using similar techniques as in McAllester (2003). $S_1$ is a batch of points (with size $m_1$) that is randomly drawn from the available set $S$ at the beginning of iteration $t$, and $S_2$ is a batch of points (with size $m_2$) that is randomly drawn from the remaining set $S \setminus S_1$. Hence, $S_1$ and $S_2$ are drawn from the set $S$ without replacement ($S_1 \cap S_2 = \emptyset$). Similar to the setting of Negrea et al. (2019); Dziugaite et al. (2020), as the random selection of indices of $S_1$ and $S_2$ is independent from the dataset $S$, $\sigma(S_1) \perp\!\!\!\perp \sigma(S_2)$, and as a result, $\mathcal{G}_1 \perp\!\!\!\perp \sigma(S_2)$ and $\mathcal{G}_2 \perp\!\!\!\perp \sigma(S_1)$. Recall that $\nu_i$ is the random parameter vector at the end of iteration $t$ that depends on $S_i$, for $i \in \{1, 2\}$. For a given sample set $S_i$, denote the conditional probability distribution of $\nu_i$ by $Q_{S_i}$. For ease of notation, we represent $Q_{S_i}$ by $Q_i$.

Let us denote

$$\Delta\left(h_{\nu_1}, h_{\nu_2}\right) \triangleq \left(L_{S_2}(h_{\nu_1}) - L(h_{\nu_1})\right) - \left(L_{S_2}(h_{\nu_2}) - L(h_{\nu_2})\right), \tag{7}$$

and

$$X_{S_2} \triangleq \sup_{Q_1, Q_2} \left(\frac{m_2}{2} - 1\right)\mathbb{E}_{\nu_1 \sim Q_1}\left[\mathbb{E}_{\nu_2 \sim Q_2}\left[\left(\Delta\left(h_{\nu_1}, h_{\nu_2}\right)\right)^2\right]\right] - \text{KL}(Q_2||Q_1). \tag{8}$$

Note that $X_{S_2}$ is a random function of the batch $S_2$. Expanding the KL-divergence, we find that

$$\left(\frac{m_2}{2} - 1\right)\mathbb{E}_{\nu_1 \sim Q_1}\left[\mathbb{E}_{\nu_2 \sim Q_2}\left[\left(\Delta\left(h_{\nu_1}, h_{\nu_2}\right)\right)^2\right]\right] - \text{KL}(Q_2||Q_1)$$

$$= \mathbb{E}_{\nu_1 \sim Q_1}\left[\left(\frac{m_2}{2} - 1\right)\mathbb{E}_{\nu_2 \sim Q_2}\left[\left(\Delta\left(h_{\nu_1}, h_{\nu_2}\right)\right)^2\right] + \mathbb{E}_{\nu_2 \sim Q_2}\left[\ln\frac{Q_1(\nu_2)}{Q_2(\nu_2)}\right]\right]$$

$$\leq \mathbb{E}_{\nu_1 \sim Q_1}\left[\ln\mathbb{E}_{\nu_2 \sim Q_2}\left[e^{(\frac{m_2}{2}-1)\left(\Delta\left(h_{\nu_1}, h_{\nu_2}\right)\right)^2}\frac{Q_1(\nu_2)}{Q_2(\nu_2)}\right]\right]$$

$$= \mathbb{E}_{\nu_1 \sim Q_1}\left[\ln\mathbb{E}_{\nu_1' \sim Q_1}\left[e^{(\frac{m_2}{2}-1)\left(\Delta\left(h_{\nu_1}, h_{\nu_1'}\right)\right)^2}\right]\right],$$

where the inequality above follows from Jensen's inequality as logarithm is a concave function. Therefore, again by applying Jensen's inequality

$$X_{S_2} \leq \ln\mathbb{E}_{\nu_1 \sim Q_1}\mathbb{E}_{\nu_1' \sim Q_1}\left[e^{(\frac{m_2}{2}-1)\left(\Delta(h_{\nu_1}, h_{\nu_1'})\right)^2}\right].$$

Taking expectations over $S_2$, we have that

$$\mathbb{E}_{S_2}\left[e^{X_{S_2}}\right] \leq \mathbb{E}_{S_2}\mathbb{E}_{\nu_1 \sim Q_1}\mathbb{E}_{\nu_1' \sim Q_1}\left[e^{(\frac{m_2}{2}-1)\left(\Delta(h_{\nu_1},h_{\nu_1'})\right)^2}\right]$$

$$= \mathbb{E}_{\nu_1 \sim Q_1}\mathbb{E}_{\nu_1' \sim Q_1}\mathbb{E}_{S_2}\left[e^{(\frac{m_2}{2}-1)\left(\Delta(h_{\nu_1},h_{\nu_1'})\right)^2}\right], \tag{9}$$

where the change of order in the expectation follows from the independence of the draw of the set $S_2$ from $\nu_1 \sim Q_1$ and $\nu_1' \sim Q_1$, i.e., $Q_1$ is $\mathcal{G}_1$-measurable and $\mathcal{G}_1 \perp\!\!\!\perp \sigma(S_2)$.

Now let

$$Z_i \triangleq l(h_{\nu_1}(x_i), y_i) - l(h_{\nu_1'}(x_i), y_i),$$

for all $1 \leq i \leq m_2$. Clearly, $Z_i \in [-1, 1]$ and because of Equations (1) and of the definition of $\Delta$ in Equation (7),

$$\Delta\left(h_{\nu_1}, h_{\nu_1'}\right) = \frac{1}{m_2}\sum_{i=1}^{m_2}\left(Z_i - \mathbb{E}[Z_i]\right).$$

Hoeffding's bound (Theorem 2) implies therefore that for any $t \geq 0$,

$$\mathbb{P}_{S_2}\left(|\Delta\left(h_{\nu_1}, h_{\nu_1'}\right)| \geq t\right) \leq 2e^{-\frac{m_2}{2}t^2}. \tag{10}$$

Denoting by $p(\Delta)$ the probability density function of $|\Delta\left(h_{\nu_1}, h_{\nu_1'}\right)|$, inequality (10) implies that for any $t \geq 0$,

$$\int_t^\infty p(\Delta)d\Delta \leq 2e^{-\frac{m_2}{2}t^2}. \tag{11}$$

The density $\tilde{p}(\Delta)$ that maximizes $\int_0^\infty e^{(\frac{m_2}{2}-1)\Delta^2}p(\Delta)d\Delta$ (the term in the first expectation of the upper bound of Equation (9)), is the density achieving equality in (11), which is $\tilde{p}(\Delta) = 2m_2\Delta e^{-\frac{m_2}{2}\Delta^2}$. As a result,

$$\mathbb{E}_{S_2}\left[e^{(\frac{m_2}{2}-1)\Delta^2}\right] \leq \int_0^\infty e^{(\frac{m_2}{2}-1)\Delta^2}2m_2\Delta e^{-\frac{m_2}{2}\Delta^2}d\Delta = \int_0^\infty 2m_2\Delta e^{-\Delta^2}d\Delta = m_2$$

and consequently, inequality (9) becomes

$$\mathbb{E}_{S_2}\left[e^{X_{S_2}}\right] \leq m_2.$$

Applying Markov's inequality on $X_{S_2}$, we have therefore that for any $0 < \delta \leq 1$,

$$\mathbb{P}_{S_2}\left[X_{S_2} \geq \ln\frac{2m_2}{\delta}\right] = \mathbb{P}_{S_2}\left[e^{X_{S_2}} \geq \frac{2m_2}{\delta}\right] \leq \frac{\delta}{2m_2}\mathbb{E}_{S_2}\left[e^{X_{S_2}}\right] \leq \frac{\delta}{2}.$$

Replacing $X_{S_2}$ by its expression defined in Equation (8), the previous inequality shows that with probability at least $1 - \delta/2$

$$\left(\frac{m_2}{2} - 1\right)\mathbb{E}_{\nu_1 \sim Q_1}\mathbb{E}_{\nu_2 \sim Q_2}\left[(\Delta(h_{\nu_1}, h_{\nu_2}))^2\right] - \mathrm{KL}(Q_2||Q_1) \leq \ln\frac{2m_2}{\delta}.$$

Using Jensen's inequality and the convexity of $(\Delta(h_{\nu_1}, h_{\nu_2}))^2$, and assuming that $m_2 > 2$, we therefore have that with probability at least $1 - \delta/2$,

$$\left(\mathbb{E}_{\nu_1 \sim Q_1}\mathbb{E}_{\nu_2 \sim Q_2}[\Delta\left(h_{\nu_1}, h_{\nu_2}\right)]\right)^2 \leq \mathbb{E}_{\nu_1 \sim Q_1}\mathbb{E}_{\nu_2 \sim Q_2}\left[(\Delta\left(h_{\nu_1}, h_{\nu_2}\right))^2\right] \leq \frac{\mathrm{KL}(Q_2||Q_1) + \ln\frac{2m_2}{\delta}}{\frac{m_2}{2} - 1}.$$

Replacing $\Delta(h_{\nu_1}, h_{\nu_2})$ by its expression Equation (7) in the above inequality, yields that with probability at least $1 - \delta/2$ over the choice of the sample set $S_2$,

$$\mathbb{E}_{\nu_1 \sim Q_1}\left[L_{S_2}(h_{\nu_1}) - L(h_{\nu_1})\right] \leq \mathbb{E}_{\nu_2 \sim Q_2}\left[L_{S_2}(h_{\nu_2}) - L(h_{\nu_2})\right] + \sqrt{\frac{2\mathrm{KL}(Q_2||Q_1) + 2\ln\frac{2m_2}{\delta}}{m_2 - 2}}.$$

$$\tag{12}$$

Similar computations with $S_1$ and $S_2$ switched, and considering that $m_1 > 2$, yields that with probability at least $1 - \delta/2$ over the choice of the sample set $S_1$,

$$\mathbb{E}_{\nu_2 \sim Q_2} [L_{S_1}(h_{\nu_2}) - L(h_{\nu_2})] \leq \mathbb{E}_{\nu_1 \sim Q_1} [L_{S_1}(h_{\nu_1}) - L(h_{\nu_1})] + \sqrt{\frac{2\mathrm{KL}(Q_1||Q_2) + 2\ln \frac{2m_1}{\delta}}{m_1 - 2}}. \tag{13}$$

The events in Equations (12) and (13) jointly hold with probability at least $1 - \delta$ over the choice of the sample sets $S_1$ and $S_2$ (using the union bound and De Morgan's law), and by adding the two inequalities we therefore have

$$\mathbb{E}_{\nu_1 \sim Q_1} [L_{S_2}(h_{\nu_1})] + \mathbb{E}_{\nu_2 \sim Q_2} [L_{S_1}(h_{\nu_2})] \leq \mathbb{E}_{\nu_2 \sim Q_2} [L_{S_2}(h_{\nu_2})] + \mathbb{E}_{\nu_1 \sim Q_1} [L_{S_1}(h_{\nu_1})]$$
$$+ \sqrt{\frac{2\mathrm{KL}(Q_2||Q_1) + 2\ln \frac{2m_2}{\delta}}{m_2 - 2}} + \sqrt{\frac{2\mathrm{KL}(Q_1||Q_2) + 2\ln \frac{2m_1}{\delta}}{m_1 - 2}},$$

which concludes the proof. □

## C  COMMON EXPERIMENTAL DETAILS

The training objective in our experiments is to minimize the cross-entropy loss, and both the cross entropy and the error percentage are displayed. The training error is computed using Equation (1) over the training set. The empirical test error also follows Equation (1) but it is computed over the test set. The generalization loss (respectively, error) is the difference between the test and the training cross entropy losses (resp., classification errors). The batch size in our experiments is 128 unless otherwise stated, the SGD learning rate is $\gamma = 0.01$ and no momentum is used (unless otherwise stated). All the experiments took at most few hours on one Nvidia Titan X Maxwell GPU. All the reported values throughout the paper are an average over at least 5 runs.

To present results throughout the training, in the x-axis of figures, both epoch and iteration are used: an epoch is the time spent to pass through the entire dataset, and an iteration is the time spent to pass through one batch of the dataset. Thus, each epoch has $B$ iterations, where $B$ is the number of batches. The convolutional neural network configurations we use are: AlexNet (Krizhevsky et al., 2012), VGG (Simonyan & Zisserman, 2014) and ResNet (He et al., 2016). In those experiments with varying width, we use a scaling factor to change both the number of channels and the number of hidden units in convolutional and fully connected layers, respectively. The default configuration is with scaling factor $= 1$.

In experiments with a random labeled training set, we modify the dataset similar to Chatterjee (2020). For a fraction of the training samples, which is the amount of noise ($0\%, 25\%, 50\%, 75\%, 100\%$), we choose the labels at random. For a classification dataset with a number $k$ of classes, if the label noise is $25\%$, then on average $75\% + 25\% * 1/k$ of the training points still have the correct label.

### C.1  RE-SCALING THE LOSS

Let us track the evolution of gradient disparity (Equation (6)) during training. As training progresses, the training losses of all the batches start to decrease when they get selected for the parameter update. Therefore, the value of gradient disparity might decrease, not necessarily because the distance between the two gradient vectors is decreasing, but because the value of each gradient vector is itself decreasing. To avoid this, a re-scaling or normalization of the loss is needed to compare gradient disparity at different stages of training. Note that this re-scaling or normalization does not affect the training algorithm, only the computation of gradient disparity.

We perform both re-scaling and normalization. The re-scaling of the loss values is given by

$$L_{S_j} = \frac{1}{m_j} \sum_{i=1}^{m_j} \frac{l_i}{\text{std}_i\left(l_i\right)},$$

where with some abuse of notation, $l_i$ is the cross entropy loss for the data point $i$ in the batch $S_j$. The normalization of the loss values is given by

$$L_{S_j} = \frac{1}{m_j} \sum_{i=1}^{m_j} \frac{l_i - \text{Min}_i\left(l_i\right)}{\text{Max}_i\left(l_i\right) - \text{Min}_i\left(l_i\right)}.$$

We experimentally compare these two ways of computing gradient disparity in Figure 5. Both the re-scaled and normalized losses, might get

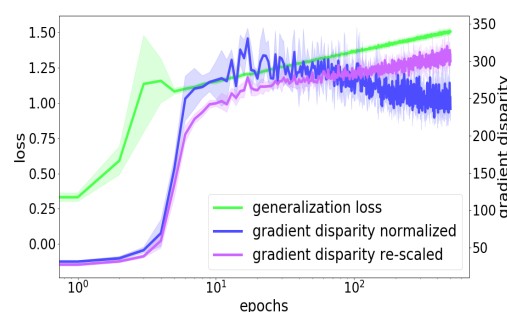

Figure 5: Normalizing versus re-scaling loss before computing average gradient disparity $\overline{\mathcal{D}}$ for a VGG-11 trained on 12.8 k points of the CIFAR-10 dataset.

unbounded if within each batch the loss values are very close to each other. However, in our experiments, we do not observe gradient disparity becoming unbounded either way. In the presence of outliers, re-scaling is more reliable than normalizing, because with normalization non-outlier data might end up in a very small interval between 0 and 1. This might explain the mismatch between the normalized gradient disparity and generalization loss at the end of training in Figure 5. Therefore, in all experiments presented in the paper, we re-scale the loss values before computing the gradient disparity[3].

## C.2 THE HYPER-PARAMETER $s$

In this section, we briefly study the choice of the size $s$ of the subset of batches to compute the average gradient disparity

$$\overline{\mathcal{D}} = \frac{1}{s(s-1)} \sum_{i=1}^{s} \sum_{j=1, j\neq i}^{s} \mathcal{D}_{i,j}.$$

Figure 6 shows the average gradient disparity when averaged over $s$ number of batches[4]. When $s = 2$, gradient disparity is the $\ell_2$ norm distance of the gradients of two randomly selected batches and has a quite high variance. Although with higher values of $s$ the results have lower variance, choosing a higher $s$ is more computationally expensive (refer to Appendix D for more details). Therefore, we find

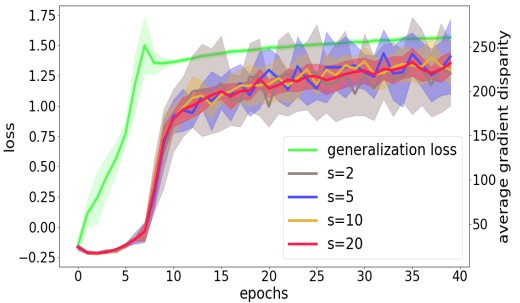

Figure 6: Average gradient disparity for different averaging parameter $s$ for a ResNet-18 that has been trained on 12.8k points of the CIFAR-10 dataset.

the choice of $s = 5$ to be sufficient enough to track down overfitting; in all the presented experiments in this paper, we use $s = 5$.

---

[3]Note that in Figure 5, both the gradient disparity and the generalization loss are increasing from the very first epoch. If we would use gradient disparity as an early stopping criterion, optimization would stop at epoch 5 and we would have a 0.36 drop in the test loss value, compared to the loss reached when the model achieves 0 training loss.

[4]In the setting of Figure 6, if using gradient disparity as an early stopping criterion, optimization would stop at epoch 9 and we would have a 0.28 drop in the test loss value compared to the loss reached when the model achieves 0 training loss.

## C.3 THE SURROGATE LOSS FUNCTION

It has been shown that cross entropy is better suited for computer-vision classification tasks compared to the mean square error (Kline & Berardi, 2005; Hui & Belkin, 2020). Hence, we choose the cross entropy criterion for all our experiments to avoid possible pitfalls of the mean square error, such as not tracking the confidence of the predictor.

Soudry et al. (2018) argues that when using cross entropy, as training proceeds, the magnitude of the network parameters increases. This can potentially affect the value of gradient disparity. Therefore, we compute the magnitude of the network parameters over iterations in various settings. We observe that this increase is very low both at the end of the training and, more importantly, at the time when gradient disparity signals overfitting (denoted by GD epoch in Table 3). Therefore, it is unlikely that the increase in the magnitude of the network parameters affects the value of gradient disparity.

Furthermore, we examine gradient disparity for models trained on the mean square error, instead of the cross entropy criterion. We observe a high correlation between gradient disparity and test error/loss (Figure 7), which is consistent with the results obtained using the cross entropy criterion. Therefore, the applicability of gradient disparity as a generalization metric is not limited to settings with the cross entropy criterion.

| Setting | at epoch 0 | at GD epoch | at epoch 200 |
|---|---|---|---|
| AlexNet, MNIST | 1 | 1.00034 | 1.00123 |
| AlexNet, MNIST, 50% random | 1 | 1.00019 | 1.00980 |
| VGG-16, CIFAR-10 | 1 | 1.00107 | 1.00127 |
| VGG-16, CIFAR-10, 50% random | 1 | 1.00222 | 1.00233 |

Table 3: The ratio of the magnitude of the network parameter vector at epoch $t$ to the magnitude of the network parameter vector at epoch 0, for $t \in \{0, \text{GD}, 200\}$, where GD stands for the epoch when gradient disparity signals to stop the training.

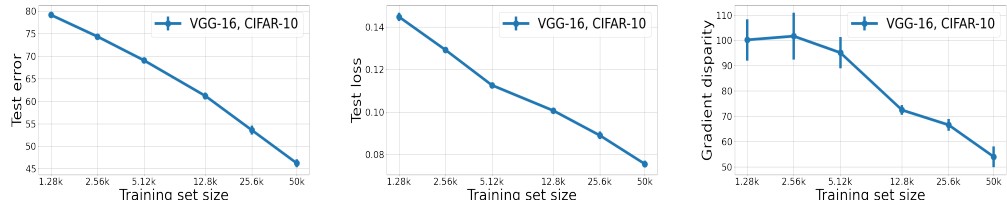

Figure 7: Test error (TE), test loss (TL), and gradient disparity ($\overline{\mathcal{D}}$) for VGG-16 trained with different training set sizes to minimize the mean square error criterion on the CIFAR-10 dataset. The Pearson correlation coefficient between TE and $\overline{\mathcal{D}}$ and between TL and $\overline{\mathcal{D}}$ are $\rho_{\overline{\mathcal{D}},\text{TE}} = 0.976$ and $\rho_{\overline{\mathcal{D}},\text{TL}} = 0.943$, respectively.

# D $k$-FOLD CROSS VALIDATION

$k$-fold cross validation is done by splitting the available dataset into $k$ sets, training on $k-1$ of them and validating on the remaining one. This is repeated $k$ times so that every set is used once as the validation set. We can adopt two different early stopping approaches: stop the optimization either (i) when the validation loss (respectively, gradient disparity) has increased for $m = 5$ epochs from the beginning of training (which is marked by the gray vertical bar in Figures 8 and 9), or (ii) when the validation loss (resp., gradient disparity) has increased for 5 consecutive epochs (which is indicated by the magenta vertical bar in Figures 8 and 9). When there is low variations and a sharp increase in the value of the metric, the two coincide (for instance, Figure 8 (b) (middle left)). In our experiments, we observe that gradient disparity appears to be less sensitive to the choice of the approaches (i) and (ii) compared to $k$-fold cross validation. Moreover, in Table 5 we study different values of $m$, which is usually referred to as *patience* among practitioners.

Early stopping should optimally occur when there is a minimum valley throughout the training in the test loss/error curves, or when the generalization loss/error starts to increase. In those experiments where such a minimum in the test loss curve exists, we compare gradient disparity to the test loss. Otherwise, we compare gradient disparity to the generalization loss. For $k$-fold cross validation, we compare validation loss to the test loss, because validation loss is expected to predict the test loss. Note that in the experiments with noisy labeled data, all the available data contains corrupted samples, hence both validation loss and gradient disparity are computed over sets which may contain corrupted samples.

**Limited Data.** We present the results for limited data scenario in Figure 8 and Table 4 for MNIST, CIFAR-10 and CIFAR-100 datasets, where we simulate the limited data scenario by using a small subset of the training set. For the CIFAR-100 experiment (Figure 8 (a) and Table 4 (top row)), we observe (from the left figure) that validation loss predicts the test loss pretty well. We observe (from the middle left figure) that gradient disparity also predicts the test loss quite well. However, the main difference between the two settings is that when using cross validation, $1/k$ of the data is set aside for validation and $1 - 1/k$ of the data is used for training. Whereas when using gradient disparity, all the data $(1 - 1/k + 1/k)$ is used for training. Hence, the test loss in the leftmost and middle left figures differ. The difference between the test accuracy (respectively, test loss) obtained in each setting is visible in the rightmost figure (resp., middle right figure). We observe that there is over $3\%$ improvement in the test accuracy when using gradient disparity as an early stopping criterion. This improvement is consistent for MNIST and CIFAR-10 datasets (Figures 8 (b) and (c) and Table 4). We conclude that in the absence of label noise, both $k$-fold cross validation and gradient disparity predict the optimal early stopping moment, but the final test loss/error is much lower for the model trained with all the available data (as when gradient disparity is used), than the model trained with a $(1 - 1/k)$ portion of the data (as in $k$-fold cross validation). To further test on a dataset that is itself limited, a medical application with limited labeled data is empirically studied later in this section (Appendix D.1). The same conclusion is made for this dataset.

**Noisy Labeled Data.** The results for datasets with noisy labels are presented in Figure 9 and Table 6 for MNIST, CIFAR-10 and CIFAR-100 datasets. We observe (from Figure 9 (a) (left)) that for the CIFAR-100 experiment, the validation loss does no longer predict the test loss. Nevertheless, although gradient disparity is computed on a training set that contains corrupted samples, it predicts the test loss quite well (Figure 9 (a) (middle left)). As a result, there is a $2\%$ improvement in the final test accuracy (for top-5 accuracy there is a $9\%$ improvement) (Table 6 (top two rows)) when using gradient disparity instead of a validation set as an early stopping criterion. This is also consistent for other configurations and datasets (Figure 9 and Table 6). We conclude that, in the presence of label noise, $k$-fold cross validation does no longer predict the test loss and fails as an early stopping criterion, unlike gradient disparity.

**Computational Cost.** Denote the time, in seconds, to compute one gradient vector, to compute the $\ell_2$ norm between two gradient vectors, to take the update step for the network parameters, and to evaluate one batch (find its validation loss and error) by $t_1$, $t_2$, $t_3$ and $t_4$, respectively. Then, one epoch of $k$-fold cross validation takes

$$\mathrm{CV_{epoch}} = k \times \left( \frac{k-1}{k} B(t_1 + t_3) + \frac{B}{k} t_4 \right)$$

seconds, where $B$ is the number of batches. Performing one epoch of training the network and computing the gradient disparity takes

$$\text{GD}_{\text{epoch}} = B(t_1 + t_3) + s\left(t_1 + \frac{s-1}{2}t_2\right)$$

seconds. In our experiments, we observe that $t_1 \approx 5.1t_2 \approx 100t_3 \approx 3.4t_4$, hence the approximate time to perform one epoch for each setting is

$$\text{CV}_{\text{epoch}} \approx (k-1)Bt_1, \qquad \text{and} \qquad \text{GD}_{\text{epoch}} \approx (B+s)t_1.$$

Therefore, as $s < B$, we have $\text{CV}_{\text{epoch}} \gg \text{GD}_{\text{epoch}}$.

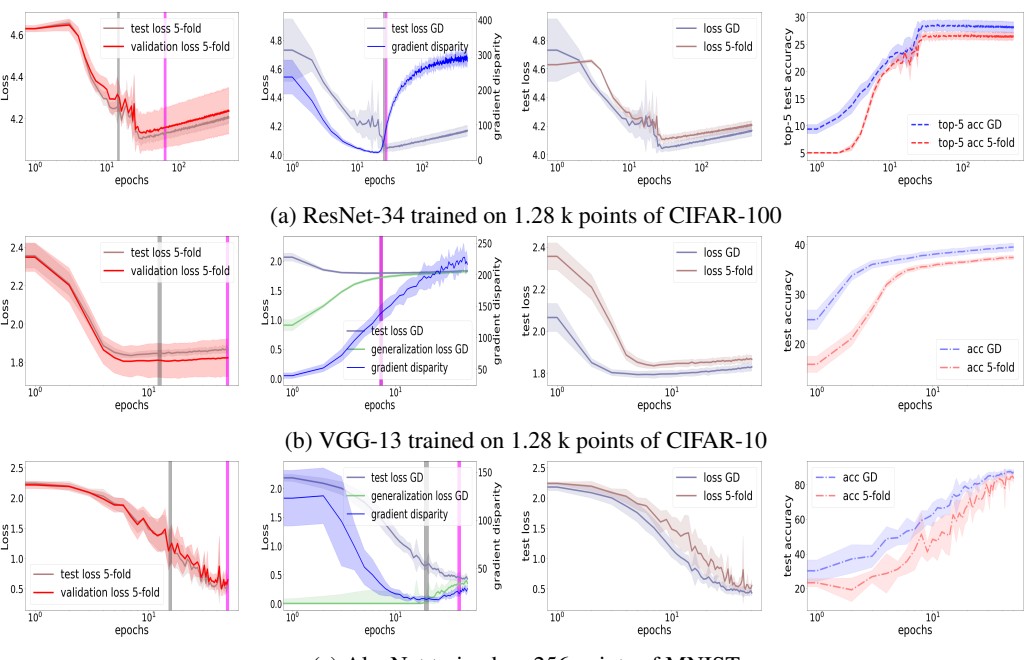

(a) ResNet-34 trained on 1.28 k points of CIFAR-100

(b) VGG-13 trained on 1.28 k points of CIFAR-10

(c) AlexNet trained on 256 points of MNIST

Figure 8: Comparing 5-fold cross validation (CV) with gradient disparity (GD) as an early stopping criterion when the available dataset is limited. (left) Validation loss versus test loss in 5-fold cross validation. (middle left) Gradient disparity versus test and generalization losses. (middle right and right) Performance on the unseen (test) data for GD versus 5-fold CV. (a) The parameters are initialized by Xavier techniques with uniform distribution. (b, c) The parameters are initialized using He technique with normal distribution. (c) The batch size is 32. The gray and magenta vertical bars indicate the epoch in which the metric (the validation loss or gradient disparity) has increased for 5 epochs from the beginning of training and for 5 consecutive epochs, respectively. In (b) the middle left figure, these two bars meet each other.

## D.1 MRNET DATASET

So far, we have presented the improvement of gradient disparity over cross validation for limited subsets of MNIST, CIFAR-10 and CIFAR-100 datasets. In this sub-section, we present the results for when the available dataset is by itself limited. We consider the MRNet dataset Bien et al. (2018) for diagnosis of knee injuries. The dataset contains 1370 magnetic resonance imaging (MRI) exams to study the presence of abnormality, anterior cruciate ligament (ACL) tears and meniscal tears. The labeled data in MRNet dataset is therefore very limited. Each MRI scan is a set of $S$ slices of images stacked together. Each patient (case) has three MRI scans: sagittal, coronal and axial. The MRNet dataset is split into training (1130 cases), validation (120 cases) and test sets (120 cases). The test set is not publicly available. We need however to set aside some data to evaluate both gradient disparity and $k$-fold cross validation, hence, in our experiments, the validation set becomes the unseen (test)

| Setting | Method | Test loss | Test accuracy |
|---|---|---|---|
| CIFAR-100, ResNet-34 | 5-fold CV | $4.249 \pm 0.028$ | $6.79 \pm 0.49$ (top-5: $22.19 \pm 0.77$) |
| | GD | $\mathbf{4.057} \pm 0.043$ | $\mathbf{9.99} \pm 0.92$ (top-5: $\mathbf{27.84} \pm 1.30$) |
| CIFAR-10, VGG-13 | 5-fold CV | $1.846 \pm 0.016$ | $35.982 \pm 0.393$ |
| | GD | $\mathbf{1.793} \pm 0.016$ | $\mathbf{36.96} \pm 0.861$ |
| MNIST, AlexNet | 5-fold CV | $1.123 \pm 0.25$ | $62.62 \pm 6.36$ |
| | GD | $\mathbf{0.656} \pm 0.080$ | $\mathbf{79.12} \pm 3.04$ |

Table 4: The loss and accuracy on the test set comparing 5-fold cross validation and gradient disparity as early stopping criterion when the available dataset is limited. The corresponding curves during training are presented in Figure 8. The above results are obtained by stopping the optimization when the metric (either validation loss or gradient disparity) has been increased for five epochs from the beginning of training.

| Patience / Method | 1 | 5 | 10 | 15 | 20 | 25 |
|---|---|---|---|---|---|---|
| 5-fold CV | $41.15 \pm 5.68$ | $62.62 \pm 6.36$ | $81.39 \pm 3.64$ | $80.39 \pm 2.88$ | $\mathbf{84.84} \pm 2.53$ | $83.55 \pm 2.84$ |
| GD | $30.19 \pm 6.21$ | $79.12 \pm 3.04$ | $84.82 \pm 2.14$ | $85.35 \pm 2.09$ | $\mathbf{87.28} \pm 1.24$ | $86.69 \pm 1.31$ |

(a) MNIST, AlexNet, limited dataset (Figure 8 (c))

| Patience / Method | 1 | 5 | 10 | 15 | 20 | 25 |
|---|---|---|---|---|---|---|
| 10-fold CV | $96.54 \pm 0.15$ | $97.28 \pm 0.20$ | $\mathbf{97.35} \pm 0.23$ | $97.22 \pm 0.19$ | $96.60 \pm 0.33$ | $94.69 \pm 0.87$ |
| GD | $97.07 \pm 0.16$ | $97.32 \pm 0.15$ | $\mathbf{97.41} \pm 0.15$ | $96.57 \pm 0.64$ | $95.44 \pm 0.96$ | $92.58 \pm 0.65$ |

(b) MNIST, AlexNet, noisy dataset (Figure 9 (d))

Table 5: The test accuracies achieved by using $k$-fold cross validation (CV) and by using gradient disparity (GD) as early stopping criteria for different patience values. For a given patience value of $m$, the training is stopped after $m$ increases in the value of the validation loss in $k$-fold CV (top rows) and of GD (bottom rows). Throughout the paper, we have chosen $m = 5$ as the default patience value for all methods without optimizing it even for GD. However, in this Table, we observe that even if we tune the patience value for $k$-fold CV and for GD separately (which is indicated in bold), GD still outperforms $k$-fold CV.

set. To perform cross validation, we split the set used for training in Bien et al. (2018) into a first subset used for training in our experiments, and a second subset used as validation set. Note that, in this dataset, because slice $S$ changes from one case to another, it is not possible to stack the data into batches, hence the batch size is 1, which may explain the fluctuations of validation loss and gradient disparity in this setting. We used the SGD optimizer with the learning rate $10^{-4}$ for training the model. Each task in this dataset is a binary classification with unbalanced set of samples and hence we report the area under the curve of the receiver operating characteristic (AUC score).

The results for three tasks: detecting ACL tears, meniscal tears and abnormality, are shown in Figure 10 and Table 7. We can observe that both the validation loss, despite a small bias, and the gradient disparity predict the generalization loss quite well. Yet, when using gradient disparity, the final test AUC score is higher (Figure 10 (right)). As mentioned earlier, for this dataset, both the validation loss and gradient disparity vary a lot. Hence, in Table 7, we present the results of early stopping, both when the metric has increased for 5 epochs from the beginning of training, and (in parenthesis) when the metric has increased for 5 consecutive epochs. We conclude that with both approaches, the use of gradient disparity as an early stopping criterion results in more than $1\%$ improvement in the test AUC score. Because the test set used in Bien et al. (2018) is not publicly available, it is not possible to compare our predictive results with Bien et al. (2018). Nevertheless, a baseline may be the results presented in `https://github.com/ahmedbesbes/mrnet`, which report a test AUC score of $88.5\%$ for the task of detecting ACL tears, whereas we observe in Table 7 that stopping training after 5 consecutive increases in gradient disparity leads to $91.52\%$

| Setting | Method | Test loss | Test accuracy |
|---|---|---|---|
| CIFAR-100, ResNet-18 | 10-fold CV | $5.023 \pm 0.083$ | $1.59 \pm 0.15$ (top-5: $6.47 \pm 0.52$) |
| | GD | $\mathbf{4.463} \pm 0.038$ | $\mathbf{3.68} \pm 0.52$ (top-5: $\mathbf{15.22} \pm 1.24$) |
| | Plug-in | $4.964 \pm 0.057$ | $1.68 \pm 0.24$ (top-5: $7.05 \pm 0.71$) |
| CIFAR-100, ResNet-34 | 10-fold CV | $4.062 \pm 0.091$ | $9.62 \pm 1.08$ (top-5: $32.06 \pm 1.47$) |
| | GD | $4.592 \pm 0.179$ | $\mathbf{10.41} \pm 1.40$ (top-5: $\mathbf{36.92} \pm 1.20$) |
| | Plug-in | $4.134 \pm 0.185$ | $10.11 \pm 1.60$ (top-5: $34.19 \pm 2.10$) |
| CIFAR-10, VGG-13 | 10-fold CV | $2.126 \pm 0.063$ | $34.88 \pm 1.66$ |
| | GD | $2.519 \pm 0.062$ | $\mathbf{36.98} \pm 0.77$ |
| | Plug-in | $2.195 \pm 0.142$ | $35.40 \pm 3.00$ |
| MNIST, AlexNet | 10-fold CV | $0.656 \pm 0.034$ | $97.28 \pm 0.20$ |
| | GD | $0.654 \pm 0.031$ | $\mathbf{97.32} \pm 0.27$ |
| | Plug-in | $0.639 \pm 0.029$ | $97.31 \pm 0.15$ |

Table 6: The loss and accuracy on the test set comparing 10-fold cross validation and gradient disparity as early stopping criterion when the available dataset is noisy. In all the experiments, 50% of the available data has random labels. The corresponding curves during training are presented in Figure 9. The above results are obtained by stopping the optimization when the metric (either validation loss or gradient disparity) has been increased for five epochs from the beginning of training. The last row in each setting, which we call plug-in, refers to when we plug-in the epoch suggested by 10-fold CV and then report the test loss and accuracy at that epoch for a network trained on the entire set. In all these settings, using GD would still result in a higher test accuracy and therefore the advantage of GD over 10-fold CV is a better characterization of overfitting.

test AUC score for this task. With further tuning, and combining the predictions found on two other MRI planes of each patient (axial and coronal), our final prediction results could even be improved.

| Task | Method | Test loss | Test AUC score |
|---|---|---|---|
| ACL | 5-fold CV | $0.973 \pm 0.111$ ($1.246 \pm 0.142$) | $79.80 \pm 1.23$ ($89.32 \pm 1.47$) |
| | GD | $\mathbf{0.842} \pm 0.101$ ($\mathbf{1.136} \pm 0.121$) | $\mathbf{81.81} \pm 1.64$ ($\mathbf{91.52} \pm 0.09$) |
| meniscal | 5-fold CV | $0.758 \pm 0.04$ ($1.163 \pm 0.127$) | $73.53 \pm 1.30$ ($72.14 \pm 0.74$) |
| | GD | $\mathbf{0.726} \pm 0.019$ ($\mathbf{1.14} \pm 0.323$) | $\mathbf{74.08} \pm 0.79$ ($\mathbf{73.80} \pm 0.24$) |
| abnormal | 5-fold CV | $0.284 \pm 0.016$ ($0.307 \pm 0.057$) | $71.016 \pm 3.66$ ($87.44 \pm 1.35$) |
| | GD | $\mathbf{0.274} \pm 0.004$ ($\mathbf{0.275} \pm 0.053$) | $\mathbf{72.67} \pm 3.85$ ($\mathbf{88.12} \pm 0.35$) |

Table 7: The loss and AUC score on the test set, comparing 5-fold cross validation to gradient disparity both as early stopping criterion for the MRNet dataset for three different tasks using the sagittal plane MRI scans. Note that an unassisted general radiologist gives on average 92%, 84% and 89% accuracy for detecting ACL tears, meniscal tears and abnormality, respectively (Bien et al., 2018). The corresponding curves during training are presented in Figure 10. We present the results of early stopping, both when the metric has increased for 5 epochs from the beginning of training, and in parenthesis when the metric has increased for 5 consecutive epochs.

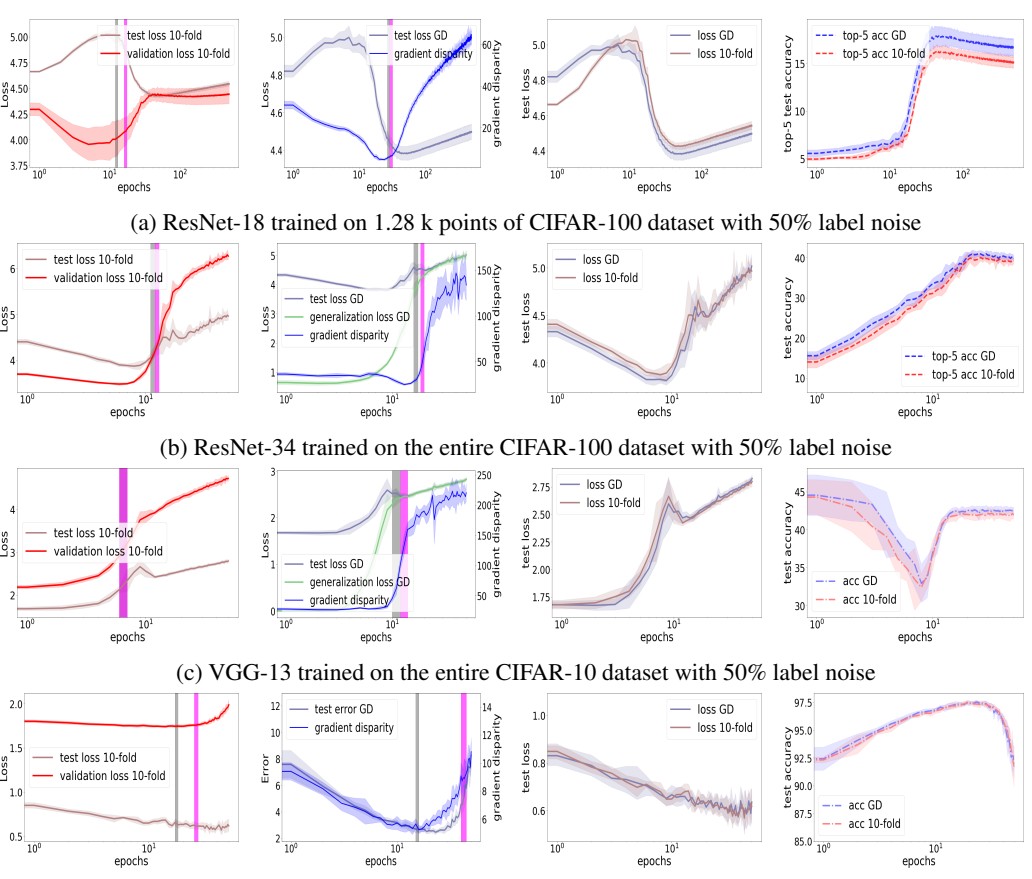

(a) ResNet-18 trained on 1.28 k points of CIFAR-100 dataset with 50% label noise

(b) ResNet-34 trained on the entire CIFAR-100 dataset with 50% label noise

(c) VGG-13 trained on the entire CIFAR-10 dataset with 50% label noise

(d) AlexNet trained on the entire MNIST dataset with 50% label noise

Figure 9: Comparing 10-fold cross validation with gradient disparity as early stopping criteria when the available dataset is noisy. (left) Validation loss versus test loss in 10-fold cross validation. (middle left) Gradient disparity versus test and generalization losses. (middle right and right) Performance on the unseen (test) data for GD versus 10-fold CV. (a) The parameters are initialized by Xavier techniques with uniform distribution. (b, c, and d) The parameters are initialized using He technique with normal distribution.

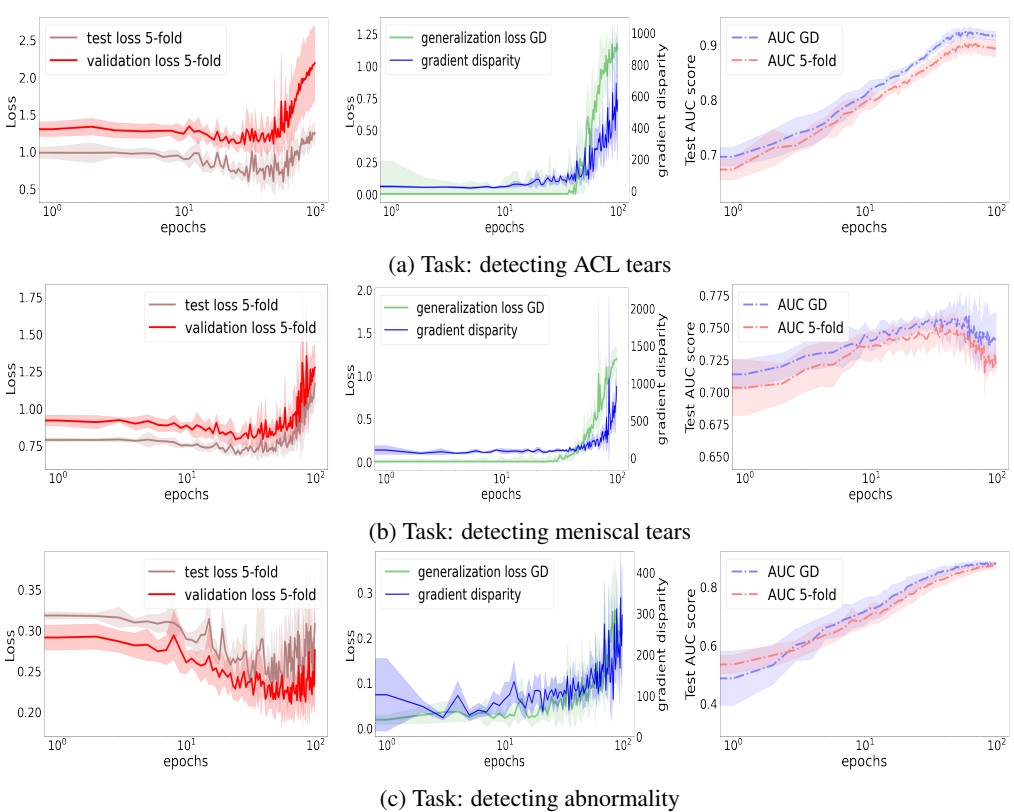

(a) Task: detecting ACL tears

(b) Task: detecting meniscal tears

(c) Task: detecting abnormality

Figure 10: Detecting three tasks from the MRNet dataset from the sagittal plane MRI scans. (left) Validation loss versus test loss in 5-fold cross validation. (middle) Gradient disparity versus generalization loss. (right) Performance comparison on the final unseen data when applying 5-fold CV versus gradient disparity.

# E    ADDITIONAL EXPERIMENTS

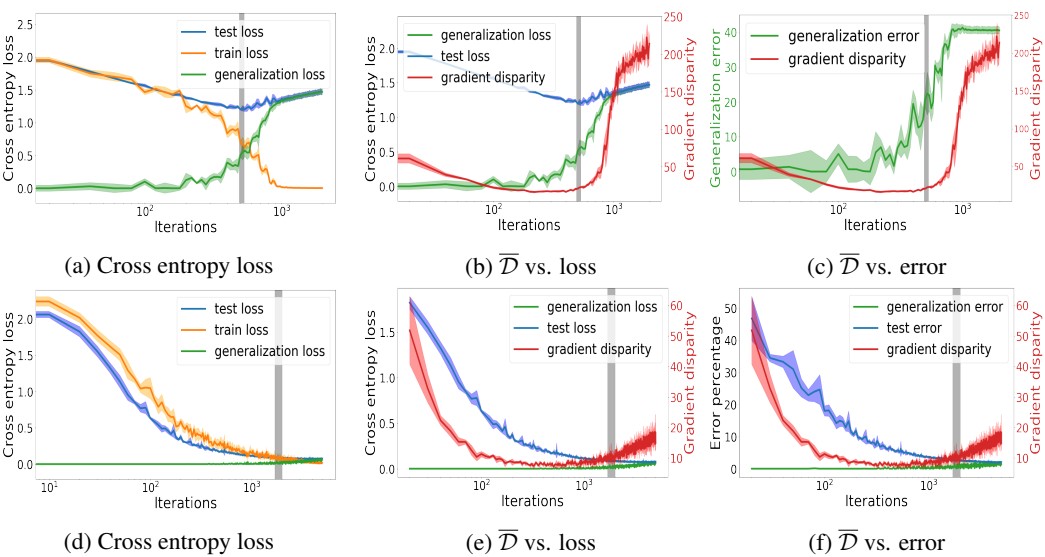

Figure 11: The cross entropy loss, the error percentage and the average gradient disparity during training. (a-c): A ResNet-18 trained on a subset of 12.8 k points of the CIFAR-10 training set (the parameter initialization is Xavier (Glorot & Bengio, 2010)). Pearson's correlation coefficient $\rho$ between $\overline{\mathcal{D}}$ and generalization loss/error over all the training iterations are $\rho_{\overline{\mathcal{D}}, \text{gen loss}} = 0.755$ and $\rho_{\overline{\mathcal{D}}, \text{gen error}} = 0.846$. (d-f): An AlexNet trained on a subset of 12.8 k points of the MNIST training set (the parameters are initialized according to the He (He et al., 2015) method with Normal distributions). For this experiment, $\rho_{\overline{\mathcal{D}}, \text{gen loss}} = 0.465$ and $\rho_{\overline{\mathcal{D}}, \text{gen error}} = 0.457$. The blue, orange, green, and red curves are the test loss/error, train loss/error, generalization loss/error, and the average gradient disparity $\overline{\mathcal{D}}$, respectively.

To investigate the relation between the average gradient disparity $\overline{\mathcal{D}}$ and generalization, we compare two sets of experiments. The first one exhibits clear overfitting, whereas in the second one, the model generalizes quite well. To test gradient disparity as a generalization metric, not only should it have very different values for each of these two sets of experiments, but it should also be well aligned with the generalization error. This is indeed what the experiments show.

In the first set of experiments, the network configuration is ResNet-18 (He et al., 2016) and it is trained on the CIFAR-10 dataset (Figure 11 (top)). Around iteration 500 (which is indicated by a thick gray vertical bar in the figures), the training and test losses (and errors) start to diverge, and the test loss reaches its minimum. This should be the early stopping point as the model is starting to overfit. Interestingly, around the same time (indicated in Figures 11 (b) and (c)), we observe a sharp increase in $\overline{\mathcal{D}}$.

The second set of experiments is on an AlexNet (Krizhevsky et al., 2012) trained on the MNIST dataset (Figure 11 (bottom)). This model generalizes quite well for this dataset. We observe that, throughout the training, the test curves are even below the training curves, which is due to the dropout regularization technique (Srivastava et al., 2014) being applied during training and not during testing. The generalization loss/error is almost zero, until around iteration 1100 (indicated in the figure by the gray vertical bar), which is when overfitting starts and the generalization error becomes non-zero. At approximately the same time, the average gradient disparity (Figures 11 (e) and (f)) starts to slightly increase, but much more slowly compared to Figures 11 (b) and (c). In both these experiments, we observe that as overfitting starts, the gradient vectors start to diverge, resulting in larger gradient disparity. These observations suggest gradient disparity as an effective early stopping criterion, as it is well aligned with the generalization error.

### E.1 MNIST EXPERIMENTS

Figure 12 shows the results for a 4-layer fully connected neural network trained on the entire MNIST training set[5]. Figures 12 (e) and (f) show the generalization losses. We observe that at the early stages of training, generalization losses do not distinguish between different label noise levels, whereas gradient disparity (Figures 12 (g) and (h)) does so from the beginning. At the middle stages of training we can observe that, surprisingly in this setting, the network with 0% label noise has higher generalization loss than the networks trained with 25%, 50% and 75%, and this is also captured by average gradient disparity. The final gradient disparity values for the networks trained with higher label noise level are also larger. For the network trained with 0% label noise we present the results with more detail in Figure 13 and observe again how gradient disparity is well aligned with the generalization loss/error. In this experiment, the early stopping time suggested by gradient disparity is epoch 9, which is the exact same time when the training and test losses/errors start to diverge, which signals therefore the start of overfitting.

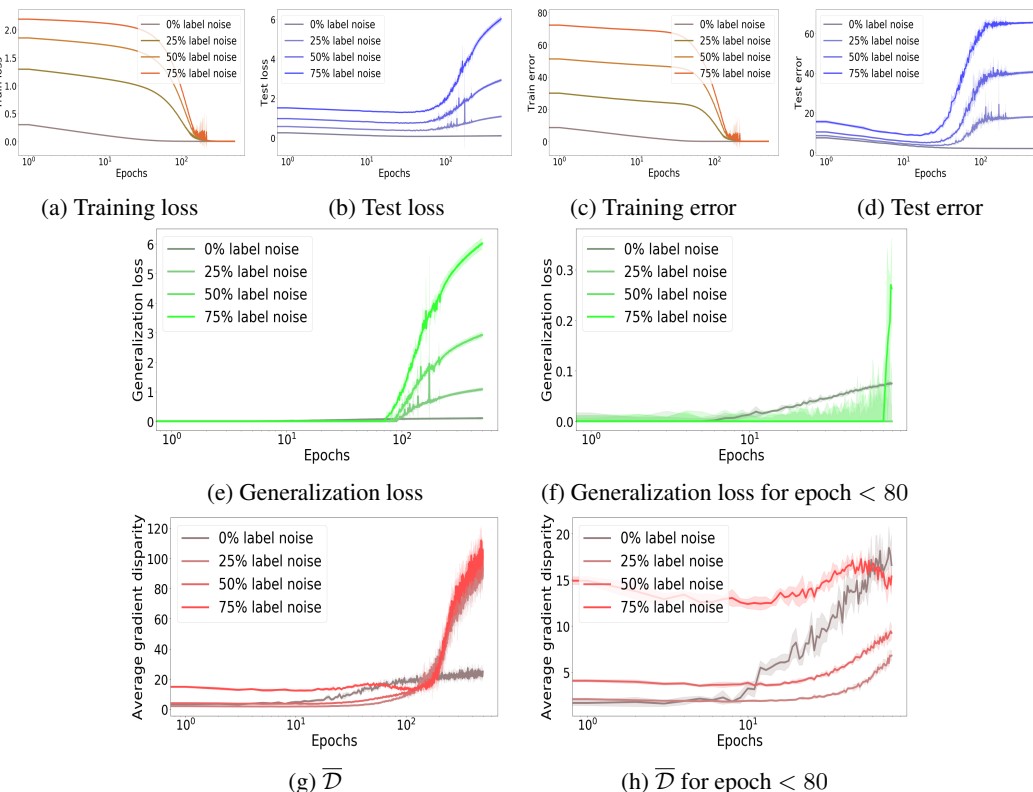

Figure 12: The cross entropy loss, error percentage, and average gradient disparity during training with different amounts of randomness in the training labels for a 4-layer fully connected neural network with 500 hidden units trained on the entire MNIST dataset. The parameter initialization is the He initialization with normal distribution.

---

[5]http://yann.lecun.com/exdb/mnist/

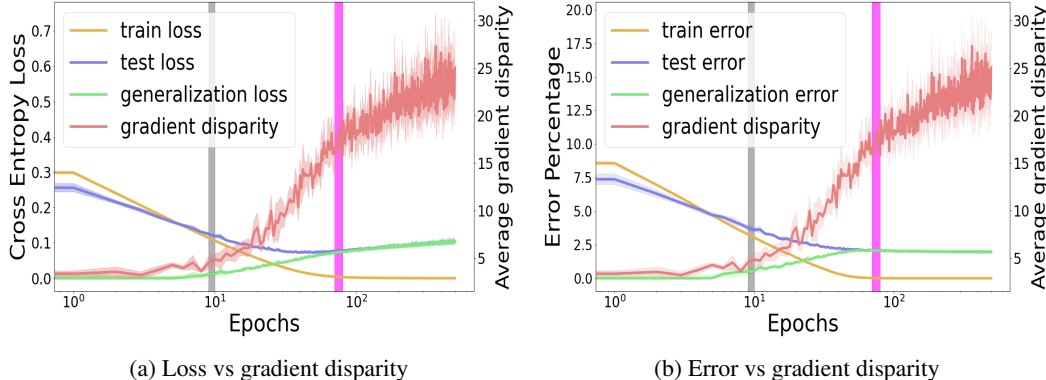

(a) Loss vs gradient disparity          (b) Error vs gradient disparity

Figure 13: The cross entropy loss, error percentage, and average gradient disparity during training for a 4-layer fully connected neural network with 500 hidden units trained on the entire MNIST dataset with 0% label noise. The parameter initialization is the He initialization with normal distribution. Pearson's correlation coefficient $\rho$ between $\overline{\mathcal{D}}$ and generalization loss/error over all the training iterations are $\rho_{\overline{\mathcal{D}},\text{gen loss}} = 0.967$ and $\rho_{\overline{\mathcal{D}},\text{gen error}} = 0.734$. The gray vertical bar indicates when GD increases for 5 epochs from the beginning of training. The magenta vertical bar indicates when GD increases for 5 *consecutive* epochs. We observe that the gray bar signals when overfitting is starting, which is when the training and testing curves are starting to diverge. The magenta bar would be a good stopping time, because if we train beyond this point, although the test error remains the same, the test loss would increase, which would result in overconfidence on wrong predictions.

### E.2 CIFAR-10 EXPERIMENTS

**Width.** To compare models with a different number of parameters using gradient disparity, we need to normalize it. The dimension of a gradient vector is the number $d$ of parameters of the model. Gradient disparity being the $\ell_2$-norm of the difference of gradient vectors will thus grow proportionally to $\sqrt{d}$, hence to compare different architectures, we propose to use the normalized gradient disparity $\tilde{\mathcal{D}} = \overline{\mathcal{D}}/\sqrt{d}$. We observe in Figure 14 that both the normalized[6] gradient disparity and test error decrease with the network width (the *scale* is a hyper-parameter used to change both the number of channels and hidden units in each configuration).

---

[6]Note that the normalization with respect to the number of parameters is different than the normalization mentioned in Section C.1 which was with respect to the loss values. The value of gradient disparity reported everywhere is the re-scaled gradient disparity; further if comparison between two different architectures is taking place the normalization with respect to dimensionality will also take place.

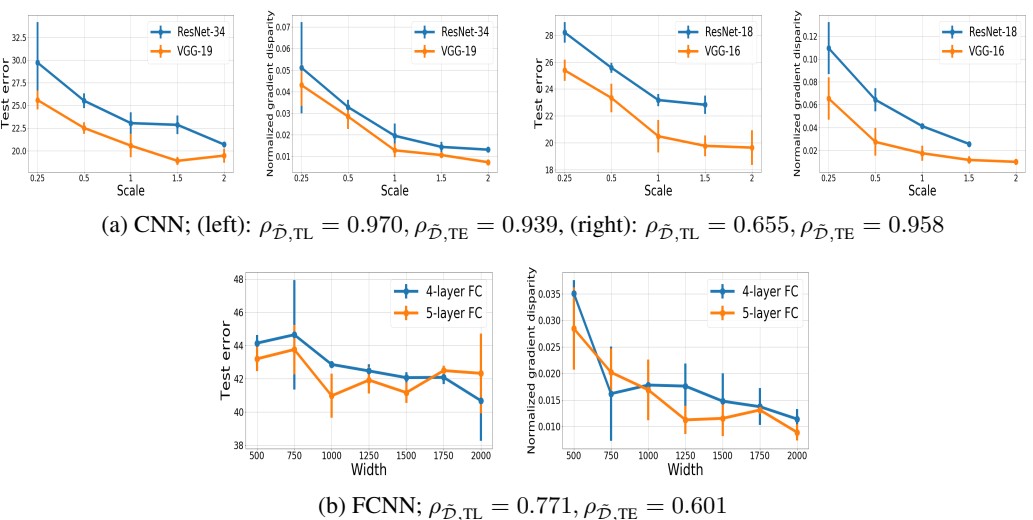

(a) CNN; (left): $\rho_{\tilde{\mathcal{D}},\text{TL}} = 0.970, \rho_{\tilde{\mathcal{D}},\text{TE}} = 0.939$, (right): $\rho_{\tilde{\mathcal{D}},\text{TL}} = 0.655, \rho_{\tilde{\mathcal{D}},\text{TE}} = 0.958$

(b) FCNN; $\rho_{\tilde{\mathcal{D}},\text{TL}} = 0.771, \rho_{\tilde{\mathcal{D}},\text{TE}} = 0.601$

Figure 14: Test error and normalized gradient disparity for networks trained on the CIFAR-10 dataset with different number of channels and hidden units for convolutional neural networks (CNN) and fully connected neural networks (FCNN). The correlation between normalized gradient disparity and test loss $\rho_{\tilde{\mathcal{D}},\text{TL}}$ and between normalized gradient disparity and test error $\rho_{\tilde{\mathcal{D}},\text{TE}}$ are reported in the captions.

Figure 15 shows the results for a 4-layer fully connected neural network, which is trained on the entire CIFAR-10 training set[7]. We observe that gradient disparity reflects the test error at the early stages of training quite well. In the later stages of training we observe that the ranking of gradient disparity values for different label noise levels matches with the ranking of generalization losses and errors. In all experiments the average gradient disparity is indeed very informative about the test error. Figure 16 shows the effect of adding data augmentation on both the test error and gradient disparity. Figure 17 shows test error and gradient disparity for networks that are trained with different training set sizes. In Figure 18, we observe that, as discussed in Section 6, gradient disparity, similar to the test error, increases with the batch size for not too large batch sizes, and as expected, when the batch size is very large (512 for the CIFAR-10 experiment and 256 for the CIFAR-100 experiments) gradient disparity starts to decrease, because gradient vectors are averaged over a large batch. Note that even with such large batch sizes, gradient disparity correctly detects the early stopping time, but the value of gradient disparity can no longer be compared to the one found with other batch sizes.

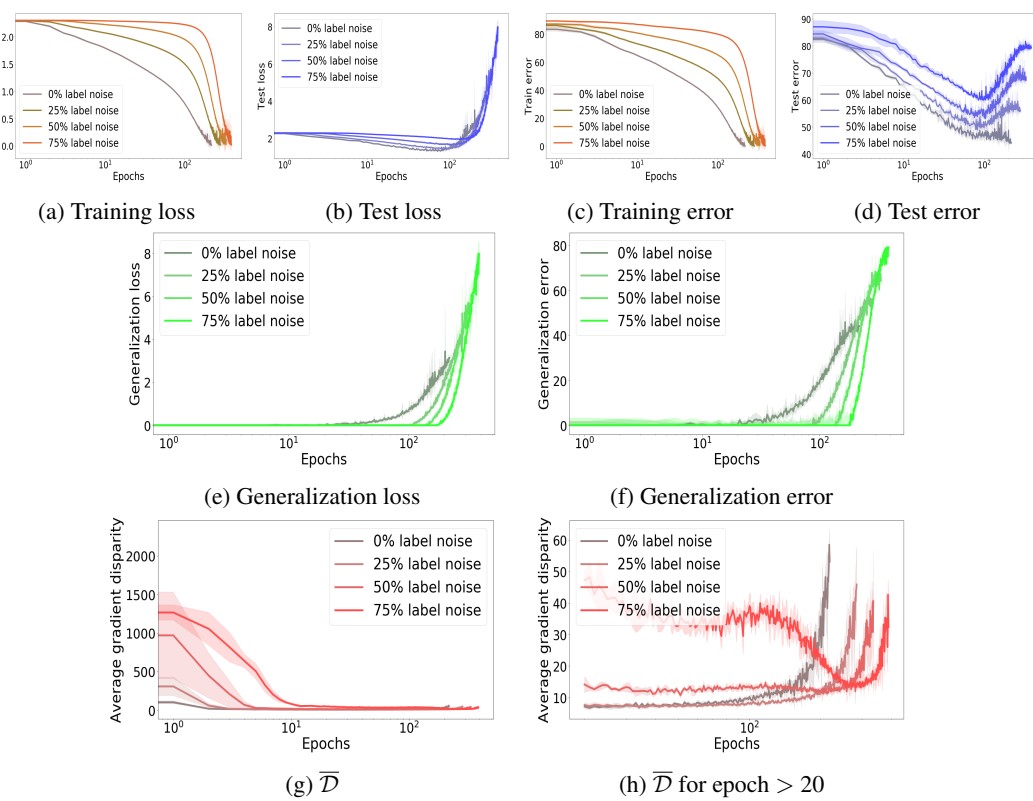

Figure 15: The cross entropy loss, error percentage, and average gradient disparity during training with different amounts of randomness in the training labels for a 4-layer fully connected neural network with 500 hidden units trained on the entire CIFAR-10 dataset. The parameter initialization is the Xavier initialization with uniform distribution. The training is stopped when the training loss gets below 0.01.

[7]https://www.cs.toronto.edu/~kriz/cifar.html

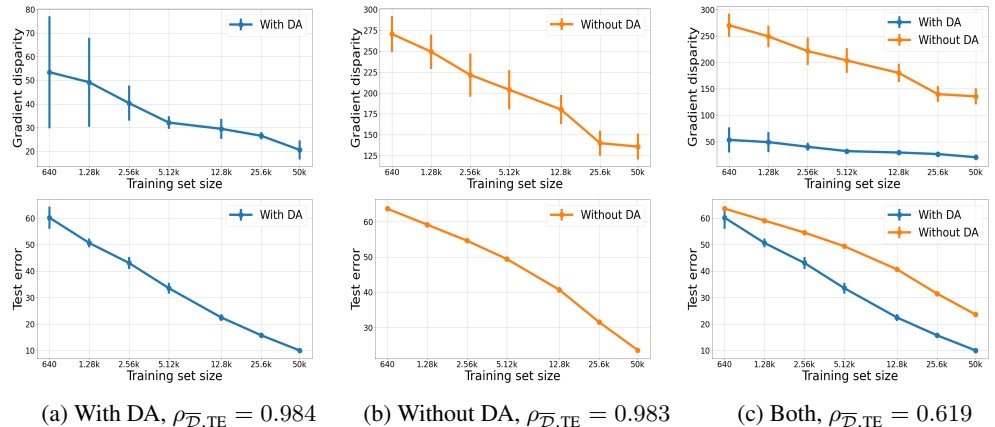

(a) With DA, $\rho_{\overline{\mathcal{D}},\mathrm{TE}} = 0.984$      (b) Without DA, $\rho_{\overline{\mathcal{D}},\mathrm{TE}} = 0.983$      (c) Both, $\rho_{\overline{\mathcal{D}},\mathrm{TE}} = 0.619$

Figure 16: The test error (bottom row) and gradient disparity (top row) for a ResNet-18 trained on the CIFAR-10 dataset with different training set sizes. (a) Results with data augmentation (DA) (we use random crop with padding $= 4$ and random horizontal flip with probability $= 0.5$). (b) Results without using any data augmentation technique. (c) Combined results of (a) and (b). We observe a strong positive correlation between gradient disparity ($\overline{\mathcal{D}}$) and test error (TE) regardless of using data augmentation or not. We also observe that using data augmentation decreases the values of both gradient disparity and the test error.

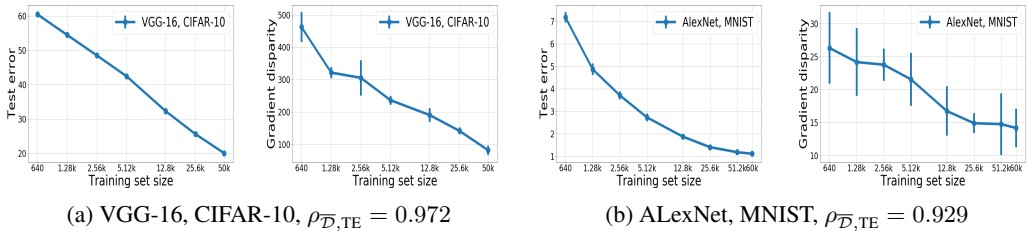

(a) VGG-16, CIFAR-10, $\rho_{\overline{\mathcal{D}},\mathrm{TE}} = 0.972$          (b) ALexNet, MNIST, $\rho_{\overline{\mathcal{D}},\mathrm{TE}} = 0.929$

Figure 17: Test error and gradient disparity for networks that are trained with different training set sizes. The training is stopped when the training loss is below $0.01$.

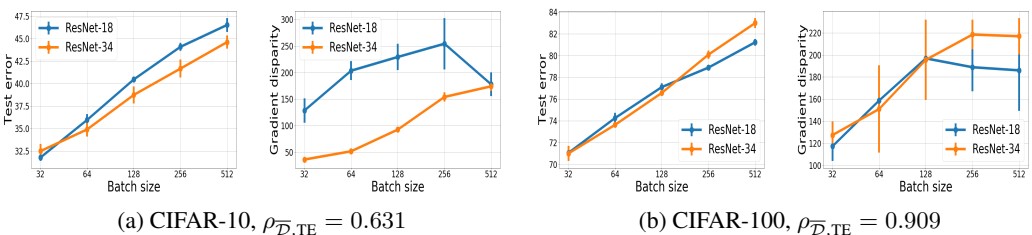

(a) CIFAR-10, $\rho_{\overline{\mathcal{D}},\mathrm{TE}} = 0.631$          (b) CIFAR-100, $\rho_{\overline{\mathcal{D}},\mathrm{TE}} = 0.909$

Figure 18: Test error and gradient disparity for networks that are trained with different batch sizes trained on $12.8$ k points of the CIFAR-10 and CIFAR-100 datasets. The training is stopped when the training loss is below $0.01$.

### E.3 CIFAR-100 EXPERIMENTS

Figure 19 shows the results for a ResNet-18 that is trained on the CIFAR-100 training set[8]. Clearly, the model is not sufficient to learn the complexity of the CIFAR-100 dataset: It has $99\%$ error for the network with $0\%$ label noise, as if it had not learned anything about the dataset and is just making a random guess for classification (because there are 100 classes, random guessing would give $99\%$ error on average). We observe from Figure 19 (f) that as training progresses, the network overfits more, and the generalization error increases. Although the test error is high (above $90\%$), very surprisingly for this example, the networks with higher label noise level, have a lower test loss and error (Figures 19 (b) and (d)). Quite interestingly gradient disparity (Figure 19 (g)) captures also this surprising trend as well.

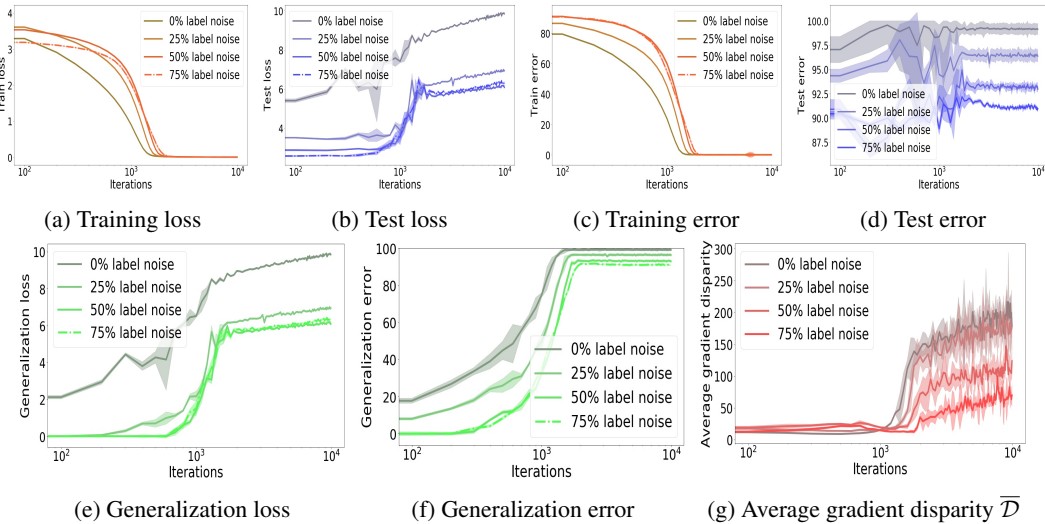

(a) Training loss     (b) Test loss     (c) Training error     (d) Test error

(e) Generalization loss     (f) Generalization error     (g) Average gradient disparity $\overline{\mathcal{D}}$

Figure 19: The cross entropy loss, error percentage, and average gradient disparity during training with different amounts of randomness in the training labels for a ResNet-18 trained on the CIFAR-100 training set. The parameter initialization is the Xavier initialization.

## F NUMBER OF BATCHES WITH LOW GRADIENT DISPARITY

In this paper, the upper bound in Theorem 1 is tracked by computing the average gradient disparity $\overline{\mathcal{D}}$ over a subset of batches. In this section, to gain some finer-grain signal of overfitting during the training process, we track down the distribution of this bound by studying the distribution of the gradient disparity, i.e., $\mathbb{P}(\mathcal{D}_{i,j} < \zeta)$ for some threshold $\zeta$.

We find the number of pairs of batches in the training set, denoted by $\mathcal{T}_\zeta$, whose gradient disparity is below the given threshold $\zeta$. For these pairs of batches, the upper bound in Theorem 1, with probability at least $1 - \delta$ and for two batches with the same size $m$, is below $2\sqrt{(\gamma^2\zeta^2/\sigma^2 + 2\ln(2m/\delta))/(m-2)}$. The lower $\mathcal{T}_\zeta$ is, the higher the average upper bound is, hence the more likely overfitting becomes. As before, for the sake of computational tractability, instead of going through all the possible pairs, we only compare $s(s-1)$ pairs of batches ($s = 5$ for our experiments), so $0 \le \mathcal{T}_\zeta \le s(s-1)$. We empirically estimate $\mathbb{P}(\mathcal{D}_{i,j} < \zeta)$ over $s$ batches by $\mathcal{T}_\zeta/(s(s-1))$. In our experiments, we compute $\mathcal{T}_\zeta$ for $\zeta \in \{10, 20\}$.

We show that as an early stopping criterion, $\mathcal{T}_\zeta$ is sometimes (slightly) more accurate than $\overline{\mathcal{D}}$. For instance, in Figure 20 (d) we highlighted in gray the minima valley of the test error for the network trained with $25\%$ noise and we observe that this aligns with the drop in $\mathcal{T}_\zeta$ for $\zeta = 20$ (light pink curve in Figure 20 (g)) better than the increasing time of $\overline{\mathcal{D}}$ (Figure 20 (f)). Also, the slow increase in the generalization loss for the AlexNet (green curve in Figure 21 (a)) is captured by the drop in

---

[8] https://www.cs.toronto.edu/~kriz/cifar.html

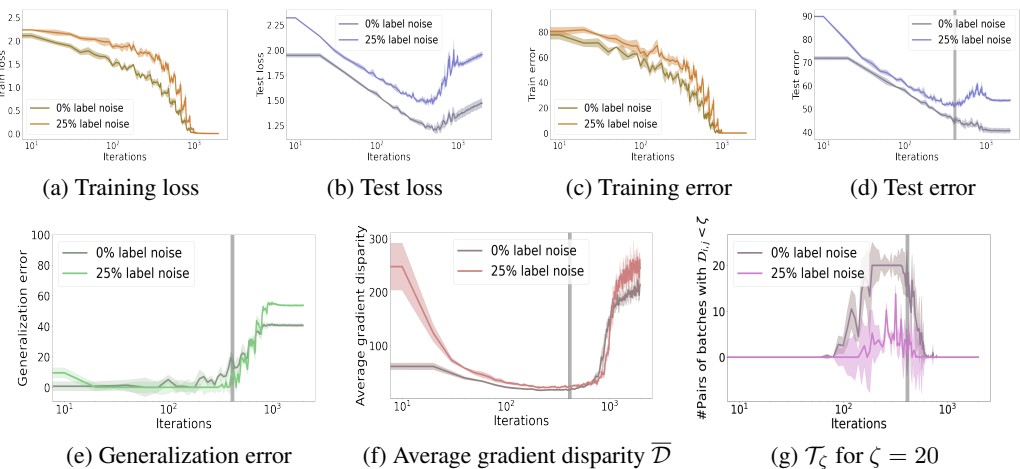

(a) Training loss     (b) Test loss     (c) Training error     (d) Test error

(e) Generalization error     (f) Average gradient disparity $\overline{\mathcal{D}}$     (g) $\mathcal{T}_\zeta$ for $\zeta = 20$

Figure 20: The cross entropy loss, error percentage, average gradient disparity and $\mathcal{T}_\zeta$ for $\zeta = 20$ during training with different amounts of randomness in the training labels for a ResNet-18 trained on a subset of 12.8 k points of the CIFAR-10 training set. We can observe that, in this setting, average gradient disparity distinguishes different label noise levels from the beginning of training, unlike generalization error.

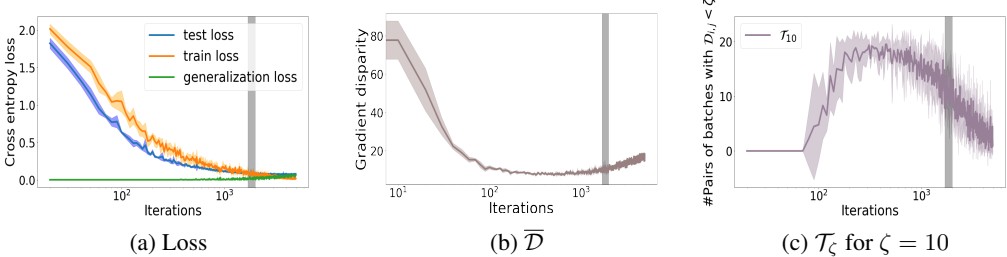

(a) Loss     (b) $\overline{\mathcal{D}}$     (c) $\mathcal{T}_\zeta$ for $\zeta = 10$

Figure 21: The cross entropy loss, average gradient disparity $\overline{\mathcal{D}}$, and $\mathcal{T}_\zeta$ for $\zeta = 10$ during training for an AlexNet trained on a subset of 12.8 k points of the MNIST training set.

$\mathcal{T}_\zeta$ for $\zeta = 10$ (Figure 21 (c)) slightly better than the increase in the average gradient disparity (Figure 21 (b)). The drawback of $\mathcal{T}_\zeta$ compared to $\overline{\mathcal{D}}$ is that it introduces an additional hyper-parameter $\zeta$ which requires tuning. In our experiments, as a rough rule of thumb, we observe that setting $\zeta$ around $10\%$ of the initial gradient disparity works well. Further empirical investigation on $\mathcal{T}_\zeta$ is left as future work.

## G  BEYOND SGD

In the following, we discuss how the analysis of Section 4 can be extended for other optimizers (refer to Ruder (2016) for an overview on popular optimizers).

### G.1  SGD WITH MOMENTUM

The momentum method (Qian, 1999) is a variation of SGD which adds a fraction of the update vector of the previous step to the current update vector to accelerate SGD:

$$v^{(t+1)} = \eta v^{(t)} + \gamma g^{(t)},$$
$$w^{(t+1)} = w^{(t)} - v^{(t+1)},$$

where $g^{(t)}$ is either $g_1$ or $g_2$ depending on the selection of the batch $S_1$ or $S_2$ for the current update step. As $\upsilon^{(t)}$ remains the same for either choice, the KL-divergence between $Q_1$ and $Q_2$ for SGD with momentum, is the same as Equation (5).

## G.2 ADAGRAD

Adagrad (Duchi et al., 2011) performs update steps with a different learning rate for each individual parameter. By denoting each coordinate of the parameter vector $w$ by $d$, one update step of the Adagrad algorithm is

$$w_d^{(t+1)} = w_d^{(t)} - \frac{\gamma}{\sqrt{G_{dd}^{(t)} + \epsilon}} g_d^{(t)}, \tag{14}$$

where the vector $g^{(t)}$ is either $g_1$ or $g_2$ depending on the selection of the batch for the current update step, and $G_{dd}^{(t)}$ is the accumulative squared norm of the gradients up until iteration $t$. Hence, for Adagrad, Equation (5) is replaced by

$$\mathrm{KL}(Q_1\|Q_2) = \frac{1}{2}\frac{\gamma^2}{\sigma^2}\left\|\frac{1}{G^{(t)} + \epsilon} \odot (g_1 - g_2)\right\|_2^2 \leq \frac{1}{2}\frac{\gamma^2}{\sigma^2}\left\|\frac{1}{G^{(t)} + \epsilon}\right\|_2^2 \|g_1 - g_2\|_2^2, \tag{15}$$

where $\odot$ denotes the element-wise product of two vectors, where division is also taken element-wise and where $\epsilon$ is a small positive constant that avoids a possible division by $0$. To compare the upper bound in Theorem 1 from one iteration to the next one (as needed to determine the early stopping moment in Section 5), gradient disparity is not the only factor in Equation (15) that evolves over time. Indeed $G^{(t)}$ is an increasing function of $t$. However, after a few iterations when the gradients become small, this value becomes approximately constant (the initial gradient values dominate the sum in $G^{(t)}$). Then the right hand side of Equation (15) varies mostly as a function of gradient disparity, and therefore gradient disparity approximately tracks down the generalization penalty upper bound.

## G.3 ADADELTA AND RMSPROP

Adadelta (Zeiler, 2012) is an extension of Adagrad, which computes a decaying average of the past gradient vectors instead of the accumulative squared norm of the gradients during the previous update steps. $G_{dd}^{(t)}$ in Equation (14) is then replaced by $\upsilon_d^{(t+1)}$ where $\upsilon_d^{(t+1)} = \eta \upsilon_d^{(t)} + (1 - \eta)(g_d^{(t)})^2$. As training proceeds, the gradient magnitude decreases. Also, $\eta$ is usually close to 1. Therefore, the dominant term in $\upsilon_d^{(t+1)}$ becomes $\eta \upsilon_d^{(t)}$. Then, if we approximate $\upsilon_1^{(t+1)} = \eta \upsilon^{(t)} + (1 - \eta)(g_1)^2 \approx \eta \upsilon^{(t)} + (1 - \eta)(g_2)^2 = \upsilon_2^{(t+1)}$ (squares are done element-wise), then for Adadelta we have

$$\mathrm{KL}(Q_1\|Q_2) \leq \frac{1}{2}\frac{\gamma^2}{\sigma^2}\left\|\frac{1}{\upsilon^{(t+1)} + \epsilon}\right\|_2^2 \|g_1 - g_2\|_2^2, \tag{16}$$

where again the division is done element-wise. The denominator in Equation (16) is smaller than the denominator in Equation (15). In both equations, the first non-constant factor in the upper bound of $\mathrm{KL}(Q_1\|Q_2)$ decreases as a function of $t$, and therefore an increase in the value of $\mathrm{KL}(Q_1\|Q_2)$ should be accounted for by an increase in the value of gradient disparity. Moreover, as training proceeds, gradient magnitudes decrease and the first factor on the upper bound of Equations (15) and (16) becomes closer to a constant. Therefore, an upper bound on the generalization penalties can be tracked by gradient disparity.

The update rule of RmsProp[9] is very similar to Adadelta, and the same conclusions can be made.

---

[9]https://www.cs.toronto.edu/~tijmen/csc321/slides/lecture_slides_lec6.pdf

### G.4 ADAM

Adam (Kingma & Ba, 2014) combines Adadelta and momentum by storing an exponentially decaying average of the previous gradients and squared gradients:

$$m^{(t+1)} = \beta_1 m^{(t)} + (1 - \beta_1)g^{(t)}, \qquad v^{(t+1)} = \beta_2 v^{(t)} + (1 - \beta_2)\left(g^{(t)}\right)^2,$$

$$\hat{m}^{(t+1)} = \frac{m^{(t+1)}}{1 - (\beta_1)^t}, \qquad \hat{v}^{(t+1)} = \frac{v^{(t+1)}}{1 - (\beta_2)^t}, \qquad w^{(t+1)} = w^{(t)} - \frac{\gamma}{\sqrt{\hat{v}^{(t+1)}} + \epsilon}\hat{m}^{(t+1)}.$$

All the operations in the above equations are done element-wise. As $\beta_2$ is usually very close to 1 (around 0.999), and as squared gradient vectors at the current update step are much smaller than the accumulated values during the previous steps, we approximate: $v_1^{(t+1)} = \beta_2 v^{(t)} + (1 - \beta_2)(g_1)^2 \approx \beta_2 v^{(t)} + (1 - \beta_2)(g_2)^2 = v_2^{(t+1)}$ (squares are done element-wise). Hence, Equation (5) becomes

$$\mathrm{KL}(Q_1||Q_2) \le \frac{1}{2}\frac{\gamma^2}{\sigma^2}\frac{1 - \beta_1}{1 - (\beta_1)^t}\left\|\frac{1}{\sqrt{\hat{v}^{(t+1)}} + \epsilon}\right\|_2^2 \|g_1 - g_2\|_2^2. \tag{17}$$

The first non-constant factor in equation above decreases with $t$ (because $\beta_1 < 1$). However it is not clear how the second factor varies as training proceeds. Therefore, unlike previous optimizers, it is more hazardous to claim that the factors other than gradient disparity in Equation (17) become constant as training proceeds. Hence, tracking only gradient disparity for the Adam optimizer may be insufficient. This is empirically investigated in the next sub-section.

### G.5 EXPERIMENTS

Figure 22 shows gradient disparity and test loss curves during the course of training for adaptive optimizers. The epoch in which the fifth increase in the value of the test loss and gradient disparity has happened is presented in the caption of each experiment. We observe that the two suggested epochs for stopping the optimization (the one suggested by gradient disparity (GD) and the other one suggested by test loss) are extremely close to each other except in Figure 22 (c) where the fifth epoch with an increase in the value of gradient disparity is much later than the epoch with the fifth increase in the value of test loss. However, in this experiment, there is a 23% improvement in the test accuracy if the optimization is stopped according to GD compared to test loss, due to many variations of test loss compared to gradient disparity.

As an early stopping criterion, the increase in the value of gradient disparity coincides with the increase in the test loss in all our experiments presented in Figure 22. In Figure 22 (h), for the Adam optimizer, we observe that after around 20 epochs, the value of gradient disparity starts to decrease, whereas the test loss continues to increase. This mismatch between test loss and gradient disparity might be due to the effect of the other factors that appear in Equation (17). Nevertheless, even in this experiment, the increase in the test loss and the gradient disparity coincide, and hence gradient disparity can correctly detect early stopping time. These experiments are a first indication that gradient disparity can be used as an early stopping criterion for optimizers other than SGD.

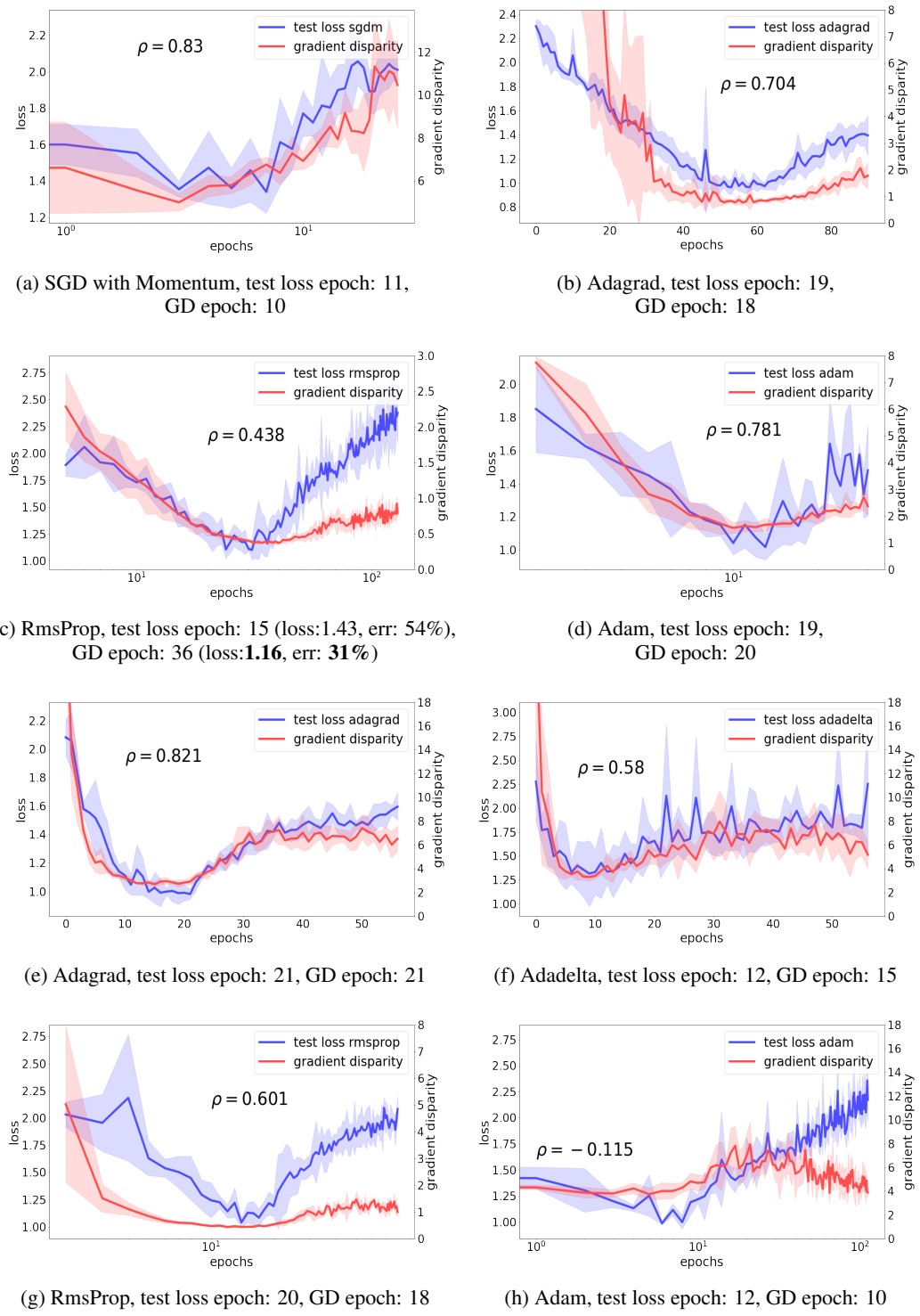

(a) SGD with Momentum, test loss epoch: 11,
GD epoch: 10

(b) Adagrad, test loss epoch: 19,
GD epoch: 18

(c) RmsProp, test loss epoch: 15 (loss:1.43, err: 54%),
GD epoch: 36 (loss:**1.16**, err: **31%**)

(d) Adam, test loss epoch: 19,
GD epoch: 20

(e) Adagrad, test loss epoch: 21, GD epoch: 21

(f) Adadelta, test loss epoch: 12, GD epoch: 15

(g) RmsProp, test loss epoch: 20, GD epoch: 18

(h) Adam, test loss epoch: 12, GD epoch: 10

Figure 22: (a-d) VGG-19 configuration trained on 12.8 k training points of CIFAR-10 dataset. (e-h) VGG-11 configuration trained on 12.8 k points of the CIFAR-10 dataset. The training is stopped when the training loss gets below 0.01. The presented results are an average over 5 runs. The captions below each figure give the epoch number where test loss and gradient disparity have respectively been increased for 5 epochs from the beginning of training. The Pearson correlation coefficient $\rho$ is presented in each figure between gradient disparity and test loss.

## H COMPARISON TO RELATED WORK

|  | Min | GD/Var | EB | GSNR | $g_i \cdot g_j$ | $\text{sign}(g_i \cdot g_j)$ | $\cos(g_i \cdot g_j)$ | $\Omega_c$ | $k$-fold | No ES |
|---|---|---|---|---|---|---|---|---|---|---|
| TE | 4.84 | **4.84** | **4.84** | 12.82 | 22.30 | 12.82 | 18.31 | 8.30 | 4.84 | 4.96 |
| TL | 0.18 | **0.18** | **0.18** | 0.46 | 0.82 | 0.46 | 0.69 | 0.32 | 0.18 | 0.22 |

(a) MNIST, AlexNet

|  | Min | GD/Var | EB | GSNR | $g_i \cdot g_j$ | $\text{sign}(g_i \cdot g_j)$ | $\cos(g_i \cdot g_j)$ | $\Omega_c$ | $k$-fold | No ES |
|---|---|---|---|---|---|---|---|---|---|---|
| TE | 13.76 | **16.66** | 24.63 | 35.68 | 37.92 | 24.63 | 35.68 | 29.40 | 17.86 | 25.725 |
| TL | 0.75 | 1.08 | **0.86** | 1.68 | 1.82 | **0.86** | 1.68 | 1.46 | 1.09 | 0.91 |

(b) MNIST, AlexNet, 50% random

|  | Min | GD/Var | EB | GSNR | $g_i \cdot g_j$ | $\text{sign}(g_i \cdot g_j)$ | $\cos(g_i \cdot g_j)$ | $\Omega_c$ | $k$-fold | No ES |
|---|---|---|---|---|---|---|---|---|---|---|
| TE | 45.54 | **45.95** | 61.76 | 70.46 | 70.46 | 55.84 | 67.09 | 67.37 | 51.64 | 64.19 |
| TL | 1.32 | **1.45** | 1.68 | 1.92 | 1.92 | 1.52 | 1.83 | 1.85 | 1.49 | 1.98 |

(c) CIFAR-10, ResNet-18

|  | Min | GD/Var | EB | GSNR | $g_i \cdot g_j$ | $\text{sign}(g_i \cdot g_j)$ | $\cos(g_i \cdot g_j)$ | $\Omega_c$ | $k$-fold | No ES |
|---|---|---|---|---|---|---|---|---|---|---|
| TE | 59.77 | 71.97 | 73.17 | 77.08 | 75.91 | **65.80** | 75.43 | 77.71 | 72.56 | 75.96 |
| TL | 1.75 | 2.00 | 2.03 | 2.12 | 2.13 | **1.93** | 2.07 | 2.13 | 2.02 | 2.30 |

(d) CIFAR-10, ResNet-18, 50% random

Table 8: Test error (TE) and test loss (TL) achieved by using various metrics as early stopping criteria. On the leftmost column, the minimum values of TE and TL over all the iterations are reported (which is not accessible during training). The results of 5-fold cross validation are reported on the right-most column, which serve as a baseline. For each experiment, we have underlined those metrics that result in a better performance than 5-fold cross validation. We observe that gradient disparity (GD) and variance of gradients (Var) consistently outperform $k$-fold cross validation, unlike other metrics. On the rightmost column (No ES) we report the results without performing early stopping (ES) (training is continued until the training loss is below 0.01).

In Table 8 we compare gradient disparity (GD) to a number of metrics that were proposed either directly as an early stopping criterion, or as a generalization metric. For those metrics that were not originally proposed as early stopping criteria, we choose a similar method for early stopping as the one we use for gradient disparity. We consider two datasets (MNIST and CIFAR-10), and two levels of label noise (0% and 50%). Here is a list of the metrics that we compute in each setting:

1. Gradient disparity (GD) (ours): we report the error and loss values at the time when the value of GD increases for the 5th time (from the beginning of the training).

2. The EB-criterion (Mahsereci et al., 2017): we report the error and loss values when EB becomes positive.

3. Gradient signal to noise ratio (GSNR) (Liu et al., 2020): we report the error and loss values when the value of GSNR decreases for the 5th time (from the beginning of the training).

4. Gradient inner product, $g_i \cdot g_j$ (Fort et al., 2019): we report the error and loss values when the value of $g_i \cdot g_j$ decreases for the 5th time (from the beginning of the training).

5. Sign of the gradient inner product, $\text{sign}(g_i \cdot g_j)$ (Fort et al., 2019): we report the error and loss values when the value of $\text{sign}(g_i \cdot g_j)$ decreases for the 5th time (from the beginning of the training).

6. Cosine similarity between gradient vectors, $\cos(g_i \cdot g_j)$ (Fort et al., 2019): we report the error and loss values when the value of $\cos(g_i \cdot g_j)$ decreases for the 5th time (from the beginning of the training).

7. Variance of gradients (Var) (Negrea et al., 2019): we report the error and loss values when the value of Var increases for the 5th time (from the beginning of the training). Variance is computed over the same number of batches used to compute gradient disparity, in order to compare metrics given the same computational budget.

8. Average gradient alignment within the class, $\Omega_c$ (Mehta et al., 2020): we report the error and loss values when the value of $\Omega_c$ decreases for the 5th time (from the beginning of the training).

On the leftmost column of Table 8, we report the minimum values of the test error and the test loss over all the iterations, which may not necessarily coincide. For instance, in setting (c), the test error is minimized at iteration 196, whereas the test loss is minimized at iteration 126. On the rightmost column of Table 8, we report the values of the test error and the test loss when using 5-fold cross validation, which serves as a baseline.

It is interesting to observe that gradient disparity and variance of gradients produce the exact same results when used as early stopping criteria (Table 8). Moreover, these two are the only metrics that consistently outperform $k$-fold cross validation. However, in Section H.1, we observe that the correlation between gradient disparity and the test loss is in general larger than the correlation between variance of gradients and the test loss.

The EB-criterion, $\text{sign}(g_i \cdot g_j)$, and $\cos(g_i \cdot g_j)$ are metrics that perform quite well as early stopping criteria, although not as well as GD and Var. In Section H.2, we observe that these metrics are not informative of the label noise level.

## H.1 GRADIENT DISPARITY VERSUS VARIANCE OF GRADIENTS

It has been shown that generalization is related to gradient alignment experimentally in Fort et al. (2019), and to variance of gradients theoretically in Negrea et al. (2019). Gradient disparity can be viewed as bringing the two together. Indeed, one can check that $\mathbb{E}\left[\mathcal{D}_{i,j}^2\right] = 2\sigma_g^2 + 2\mu_g^T\mu_g - 2\mathbb{E}\left[g_i^T g_j\right]$, given that $\mu_g = \mathbb{E}[g_i] = \mathbb{E}[g_j]$ and $\sigma_g^2 = \text{tr}\left(\text{Cov}\left[g_i\right]\right) = \text{tr}\left(\text{Cov}\left[g_j\right]\right)$. This shows that gradient variance $\sigma_g^2$ and gradient alignment $g_i^T g_j$ both appear as components of gradient disparity. We conjecture that the dominant term in gradient disparity is the variance of gradients, hence as early stopping criteria these two metrics almost always signal overfitting simultaneously. This is indeed what our experiments show; we show that variance of gradients is also a very promising early stopping criterion (Table 8). However, because of the additional term in gradient disparity (the gradients inner product), gradient disparity emphasizes the alignment or misalignment of the gradient vectors. This could be the reason why gradient disparity in general outperforms variance of gradients in tracking the value of the generalization loss; the positive correlation between gradient disparity and the test loss is often larger than the positive correlation between variance of gradients and the test loss (Table 9).

| Setting | $\rho_{\overline{\mathcal{D}},\text{TL}}$ | $\rho_{\text{Var},\text{TL}}$ |
|---|---|---|
| AlexNet, MNIST | **0.433** | 0.169 |
| AlexNet, MNIST, $50\%$ random labels | **0.535** | 0.161 |
| VGG-16, CIFAR-10 | 0.190 | **0.324** |
| VGG-16, CIFAR-10, $50\%$ random labels | **0.634** | 0.623 |
| VGG-19, CIFAR-10 | **0.685** | 0.508 |
| VGG-19, CIFAR-10, $50\%$ random labels | **0.748** | 0.735 |
| ResNet-18, CIFAR-10 | **0.975** | 0.958 |
| ResNet-18, CIFAR-10, $50\%$ random labels | **0.471** | 0.457 |

Table 9: Pearson's correlation coefficient between gradient disparity ($\overline{\mathcal{D}}$) and test loss (TL) over the training iterations is compared to the correlation between variance of gradients (Var) and test loss.

## H.2 CAPTURING LABEL NOISE LEVEL

In this section, we show in particular three metrics that even though perform relatively well as early stopping criteria, fail to account for the level of label noise, contrary to gradient disparity.

- The sign of the gradient inner product, $\text{sign}(g_i \cdot g_j)$, should be inversely related to the test loss; it should decrease when overfitting increases. However, we observe that the value of $\text{sign}(g_i \cdot g_j)$ is larger for the setting with the higher label noise level; it incorrectly detects the setting with the higher label noise level as the setting with the better generalization performance (see Figure 23).

- The EB-criterion should be larger for settings with higher overfitting. In most stages of training, the EB-criterion does not distinguish between settings with different label noise levels, contrary to gradient disparity (see Figure 23). At the end of the training, the EB-criterion even mistakenly signals the setting with the higher label noise level as the setting with the better generalization performance.

- The cosine similarity between gradient vectors, $\cos(g_i \cdot g_j)$, should decrease when overfitting increases and therefore with the level of label noise in the training data. But $\cos(g_i \cdot g_j)$ appears not to be sensitive to the label noise level, and in some cases (Figure 24 (a)) it even increases with the noise level. Gradient disparity is much more informative of the label noise level compared to cosine similarity and the correlation between gradient disparity and the test error is larger than the correlation between cosine similarity and the test accuracy (see Figure 24).

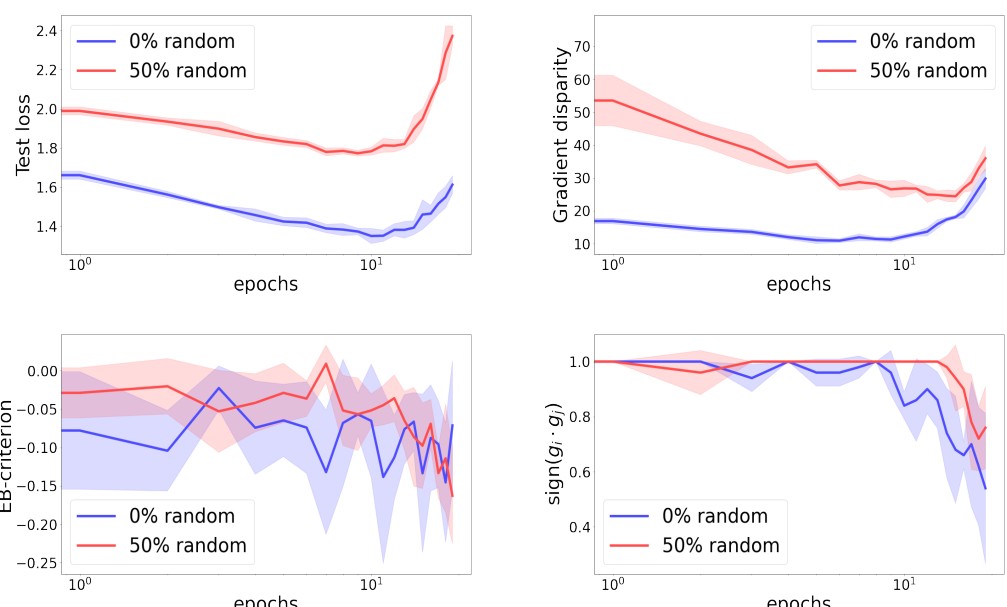

Figure 23: Test loss, gradient disparity, EB-criterion Mahsereci et al. (2017), and $\text{sign}(g_i \cdot g_j)$ for a ResNet-18 trained on the CIFAR-10 dataset, with $0\%$ and $50\%$ random labels. Gradient disparity, contrary to EB-criterion and $\text{sign}(g_i \cdot g_j)$, clearly distinguishes the setting with real labels from the setting with random labels.

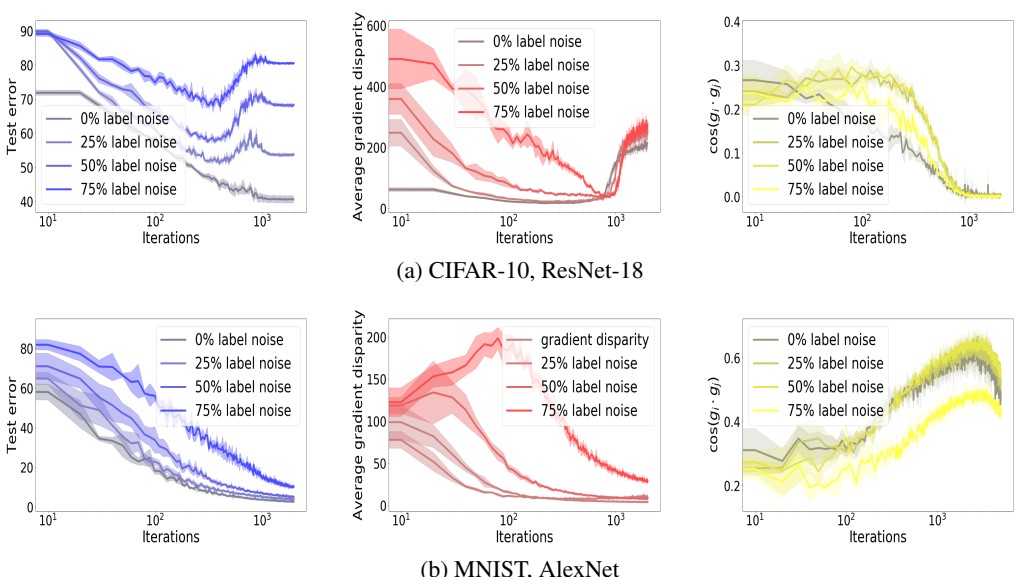

Figure 24: The test error (TE), average gradient disparity ($\overline{\mathcal{D}}$), and cosine similarity ($\cos(g_i \cdot g_j)$) during training with different amounts of randomness in the training labels for two sets of experiments. (a) ResNet-18 trained on 12.8k points of the CIFAR-10 training set. The Pearson correlation coefficient between test accuracy (TA) and cosine similarity (cos) over all levels of randomness and over all the iterations is $\rho_{\cos,\text{TA}} = -0.0088$, whereas the correlation between test error/generalization error and gradient disparity is $\rho_{\overline{\mathcal{D}},\text{TE}} = 0.2029$ and $\rho_{\overline{\mathcal{D}},\text{GE}} = 0.5268$, respectively. (b) AlexNet configuration trained on 12.8k points of the MNIST dataset. The correlation between the test accuracy and cosine similarity is $\rho_{\cos,\text{TA}} = 0.7521$, which is positive and relatively high for this experiment. Yet, it is still lower than the correlation between test error and gradient disparity which is $\rho_{\overline{\mathcal{D}},\text{TE}} = 0.8019$.

