# OpenReview forum: "Early Stopping by Gradient Disparity"
_ICLR.cc/2021/Conference — Reject_

### Official Review · AnonReviewer2 · 2020-10-28
**theoretically motivated and easy-to-compute metric for early stopping using only training set, extensive experiments**

**Rating:** 7
**Confidence:** 3

**Review:**

This paper proposes a novel early-stopping criterion called gradient disparity, which is the the l2 norm of the difference between two gradient vectors on two different batches from the training set. In contrast to typical early stopping techniques that require validation error on a held-out dataset, the proposed criterion fully operates on the training set.

The gradient disparity is theoretically motivated by an upper bound of the newly introduced concept called “generalization penalty”, which measures how much the loss on a batch of points increases due to model updating with another batch instead of itself. The upper bound takes the form of KL divergence between the two posterior model distributions conditioned on the respective batches. The posterior distributions are assumed to be Gaussian with mean being the gradient-updated weights, hence the KL divergence boils down the l2-norm difference between the gradient vectors of the two batches.

In practice, the algorithm take, say, five batches and compute the average pair-wise GD as the early-stopping criterion, where the GD is computed with the loss values of each point rescaled by the standard deviation of the loss values in the batch.

Overall I vote for accept.

Pros:

The method appears to be well motivated both theoretically and intuitively; empirically results show that the proposed gradient disparity strongly correlates with generalization error. The experiments are quite extensive.

Cons:

There are some aspects not clear to me. What exactly is your early-stopping algorithm? Do you randomly sample, e.g., five batches, from the training set after each model update, and then compute the pairwise GD and then average? Do you early-stop when the GD metric increases, or there is some kind of threshold (I suppose this metric would be noisy)?

As a sanity check, maybe add an experiment comparing early-stopping with no early-stopping?

---

> ### Author Response · Authors · 2020-11-19
> **Response to Reviewer2**
>
> Thanks for your detailed review that also summarizes the theoretical contribution of our work.
>
> The early stopping threshold that we use is as follows. At each update step, we randomly sample $s$ batches and compute the average pairwise GD (in our work $s$ is taken to be 5, but other values were also studied in Appendix C.2). We then report the results both (i) when GD (or the validation loss in $k$-fold CV) increases after $m = 5$ steps from the beginning of training (indicated by the gray vertical bar in Figures 8 and 9) and (ii) when it increases after 5 consecutive steps (indicated by the magenta vertical bar in Figures 8 and 9). We then compare using GD to $k$-fold CV. Indeed both GD and $k$-fold CV might be sensitive to the choice of (i) or (ii) (or even the number $m$ of increases, which is usually called the patience parameter among practitioners). However, we observe (e.g., in Figure 8) that GD appears to be less sensitive to the choice between (i) and (ii) compared to $k$-fold CV (the gray and magenta bars are closer for the 2nd column compared to the 1st column). Moreover, we observe that using GD is superior to $k$-fold CV with either (i) or (ii) (Table 1) and whether we optimize or not for the patience value $m$ (Table 5 of the revised paper).
>
> Following the reviewer’s suggestion, and as a sanity check, we have updated Table 8 and have reported in the rightmost column the test error and loss values if no early stopping criterion is used. Table 8 now contains a summary that compares all the baselines (please see the revised paper).

---

### Official Review · AnonReviewer4 · 2020-10-28
**A new generalization metric as an early stopping criterion**

**Rating:** 5
**Confidence:** 4

**Review:**

Summary:
The paper proposed to use the average of l_2 distance between stochastic gradients called gradient disparity as a metric to predict generalization. Early stopping is achieved by monitoring such a metric without needing a validation set. Some theoretical results are also given to motivate the use of gradient disparity as a generalization metric.

Pros:
Overall the paper is well written. The use of a generalization metric to replace the validation set is an interesting idea. From the experiments in the paper, it seems the proposed metric indeed shares a similar trend as the test error. Also, the paper compared with a few other generalization metrics in Appendix H and showed the proposed one performs better than the baseline metrics when used as early stopping criteria. The experiments look quite comprehensive to me and the effects of different factors (label noise, batch size, network width, etc) on the gradient disparity are studied.

Cons:
1. It looks like gradient disparity can be more correlated to generalization error instead of test error on some datasets (e.g., in Figure 13 for MNIST). This could make the algorithm stop too early since in some cases, the generalization error is increasing, but the training error decreases even faster and overall the test error is decreasing. However, gradient disparity has a similar trend with generalization error and when it increases for 5 epochs, we will terminate the algorithm while the test error is still decreasing. In addition, using test error + gradient disparity as a proxy for test error is not valid since gradient disparity has a different scale instead of while test error is between 0 and 1.

2. The scale of gradient disparity may change with gradient magnitude and the authors used a re-scaling heuristic to stabilize the scale. I think the metric scale is an important issue when the metric is used for early stopping. However, the effect of re-scaling is not comprehensively studied and discussed in the experiments.

---

> ### Author Response · Authors · 2020-11-19
> **Response to Reviewer4**
>
> Thanks for your review. Below we will address your two concerns:
>
> 1. What we study in this paper is the ability of GD, which is theoretically motivated by the bound in Theorem 1 and which is easy to compute (as pointed out by Reviewer2), to signal overfitting in a variety of scenarios and experimental settings. Our experiments show a strong positive correlation between GD and the generalization gap (the generalization error or loss). But more importantly, GD signals overfitting in scenarios where the generalization gap fails to signal it. For example, in Figure 2 the generalization gap does not distinguish between networks trained on different label noise levels, whereas GD does. While we propose GD as a practical signal of overfitting, we do not claim that GD predicts precisely the value of the generalization error (or of the test error) as they have different scales.
>
>    *Regarding Figure 13*: To use GD as an early stopping criterion we have provided two thresholds: stop training (i) when there are $m = 5$ (consecutive or nonconsecutive) increases in the value of GD (this number is commonly referred to as the “patience” parameter among practitioners), which is indicated by the gray vertical bars in Figures 8 and 9, or (ii) when there are 5 *consecutive* increases in the value of GD, which is indicated by the magenta vertical bars. In Figure 13, we had only indicated (i) by the gray bar. We have now added (ii) (the magenta bar) in Figure 13 as well (please see the revised paper). We observe that if training continues after (ii) (the magenta bar), the test loss value increases, and the test error remains the same. Therefore the network starts to become more overconfident in its wrong predictions. Using GD with the threshold (ii) would therefore avoid this. For this experiment, the results suggest to use threshold (ii), or alternatively to use threshold (i) with a higher patience value (see the discussion below).
>
>    *Sensitivity to the choice of the thresholds (i) or (ii)*: Which exact patience parameter to choose as an early stopping threshold, or whether it should include non-consecutive increases or not, is indeed an interesting question [1], which does not have a definite answer to date even for $k$-fold CV. In this work, we consider two possible thresholds (i) and (ii) for both GD and $k$-fold CV. In most of the experiments (more precisely, in 5 out of 7 experiments of Figures 8 and 9), GD is not sensitive to the choice of the threshold (see Figures 8(a), 8(b), 9(a), 9(b) and 9(c) on the 2nd column, where the gray and magenta bars almost coincide). In contrast, $k$-fold CV is more sensitive (see for example the leftmost column of Figures 8(a) and 8(b), where the gray and magenta bars are very far away when using $k$-fold CV). In the other 2 experiments (Figures 8(c) and 9(d), both with the MNIST dataset), the thresholds (i) and (ii) do not coincide for neither GD nor $k$-fold CV.
>
>    *Sensitivity to the patience parameter $m$*: In this work, we have chosen $m = 5$ as the default patience value for all methods without optimizing it (typical values of $m$ lie between 1 and 10). Hence it might not be the optimal value, in particular for the MNIST experiments. To study the sensitivity to the value of $m$ (patience), we have provided the test accuracy for experiments of Figures 8(c) and 9(d) (which are on the MNIST dataset), for different values of $m$ in Table 5 of the revised paper. We observe that $m = 5$ was indeed low both for GD and for $k$-fold CV. However, even if we optimize $m$ for $k$-fold CV (reported in bold in Table 5), GD still outperforms $k$-fold CV.
>
>    To sum up, what we want to emphasize in Section 5 (and Appendix D), is that GD can potentially replace $k$-fold CV, especially when data is limited and/or noisy. The optimization of the patience value, whether it is for GD or $k$-fold CV, might indeed require tuning it for the particular dataset. In the current paper, we observe however that whether we use threshold (i) or (ii), or whether we allow optimization or not for the patience value, GD leads to a better performance than $k$-fold CV (Tables 1 and 5).
>
> 2. As training progresses, the training loss values and therefore the gradient magnitudes start to decrease, which in turn affects the scale of GD and requires a re-scaling. This is why before computing GD the loss values are re-scaled within each batch. We discuss two possible options for this re-scaling in Appendix C.1: (i) re-scaling by the standard deviation or (ii) normalizing by the difference between min and max values. It is important to note that this re-scaling is only done for computing GD as a metric, and therefore does not have any effect on the training process itself (we do not perform re-scaling before applying back-propagation).
>
> [1] Lutz Prechelt. Early stopping-but when? In Neural Networks: Tricks of the trade, pp. 55–69. Springer, 1998.

---

### Official Review · AnonReviewer1 · 2020-10-29
**On the fairness of experimental comparison between CV and GD due to different sample size**

**Rating:** 5
**Confidence:** 3

**Review:**

The paper proposes using gradient disparity between two batches for early stopping, and explains the reason by showing its connection with the generalization error. Experiments on limited-size dataset and noisy dataset are presented.

My main concern is related to the experimental setting: The experiments in Section 5 compares

(a) k-CV with (1-1/k)N training samples + N/k validation samples; with

(b) Gradient Disparity (GD) using  N training samples.


Why not also compare with

(x1) Gradient Disparity (GD) using (1-1/k)N training samples; or maybe even

(x2) use k-CV to determine stopping epoch, take an average to estimate the best stopping epoch \hat{n}, then re-train using all samples and stop at epoch \hat{n}.

It seems unclear to me whether the advantage of GD comes from (i) better characterization of the generalization or (ii) sample size advantage.

(x1) v.s. (a) and (x2) v.s. (b) could show whether GD better captures the generalization under same number of samples.

(x1) v.s. (b) and (x2) v.s. (a) could show the benefit of increasing sample size.

Finally, the experimental setting uses k-fold CV instead of a fixed validation set, and it is not clear what the standard deviation of the experimental results for k-CV means, e.g., the standard deviation describes (i) randomness due to data splitting; or (ii) randomness due to training algorithm?

---

> ### Author Response · Authors · 2020-11-19
> **Response to Reviewer1**
>
> Thanks for your review. We highlight here how we compare $k$-fold CV and GD.
>
> The comparison made between $k$-fold CV and GD is fair because both algorithms are trained with exactly the same number of samples. $k$-fold CV requires to split the dataset of $N$ samples in a first subset of $(1-1/k)N$ samples and a second small set with the remaining $N/k$ samples. This splitting and cross-validation process is repeated $k$ times, so that each of the small subsets of $N/k$ samples is used exactly once for validation and $(k-1)$ times for training. All the $N$ samples are therefore used for both training and validation. In contrast, GD is directly used to indicate early stopping during the course of training, which allows us to train the network on the entire dataset of $N$ samples, by eliminating the need to set a small subset of $N/k$ samples aside and to repeat the process $k$ times to use all the samples in training and validation as in $k$-fold CV.
>
> The advantages of GD are both (i) a sample size advantage and/or (ii) a better characterization of overfitting. More precisely:
>
> (i) When the available dataset is limited (for example, for the MRNet dataset), GD allows us to use the entire dataset for training and therefore GD can achieve better test accuracy compared to $k$-fold CV (see above).
>
> (ii) When the available dataset is noisy, then $k$-fold CV might no longer be reliable (as shown for example on the leftmost column of Figure (9) (a)), contrary to GD (as shown on the second left column of Figure (9) (a)). In this case, we can perform algorithms such as the method (x2) suggested by the reviewer. We have done so and have updated Table 6 by adding the corresponding values; please see the revised paper, where the method (x2) is called “plug-in”, because we “plug-in” the epoch suggested by k-fold CV to stop training, and then report the test loss and accuracy at that epoch for a network trained on the entire set. We observe that this method (x2) gives indeed slightly higher test accuracy than $k$-fold CV, but we also see that the test accuracy achieved by GD is still higher than (x2).
>
> Finally, the standard deviation in $k$-fold CV experiments is due to the randomness of the data split. We report the average values and the standard deviation over the $k$ folds.

---

### Official Review · AnonReviewer3 · 2020-10-29
**Good idea but need improvement on explaination and exepriments**

**Rating:** 5
**Confidence:** 4

**Review:**

### After rebuttal ###
Thanks a lot for the authors' extensive experiments and good explanation to my questions. I would increase my score. The basic idea of this paper is promising and useful. However, there are still several problems after reading the rebuttal.

Figure 2 shows that GD can distinguish different noisy levels, however, it is not a very realistic setup when training with a noisy dataset, to be specific, the dataset only has one noisy lever rather than multiple. Furthermore, GD is not able to give an early-stopping criterion on noisy datasets from Figure 2 where test error keeps decreasing but GD is increasing. Including noisy datasets analysis seems not a significant contribution.

### Original comment ###
This paper aims to provide an early stopping criterion measured by gradient disparity when training with limited data.  The authors also provide theoretical insights on inducing the gradient disparity and empirically show the proposed method is robust to label noise.

However, there are several points not clear to me.
1. In the main content, comparison between CV and GD are all based on a 1.28K dataset from CIFAR100/10 and MNIST, is there an experiment showing the performance on the full dataset baseline with label noise.
2. Regarding Figure 9 in the appendix, the figure failed to explain the correlation between GD (2nd column) and test accuracy (4th column) where GD goes up. Still, accuracy has various behaviors. Is the method *only* work with a small dataset?
3. Based on Figure 3, I assume the authors do not use data augmentation in their experiment (ResNet18 has 78% accuracy on full CIFAR10); however, data augmentation is a very basic technic in model training. Will data augmentation affects the performance of GD?
4. Figure 2 is not clear to me. What is the difference between Test error and Generalization error? How do you compute them? In Figure2. (b), the scale of the Y-axis seems too large. Nothing can be observed from this figure.
5. Although the authors state the correlation \pho in their results, there's no explicit figure showing the correlation in main content like Figure 13. I believe adding them can be a good bonus.

---

> ### Author Response · Authors · 2020-11-19
> **Response to Reviewer3**
>
> Thanks for your review. We answer your questions using the same ordering as in the review:
> 1. Yes, we give results comparing CV and GD for networks that are trained on the entire MNIST, CIFAR-10, and CIFAR-100 datasets in Figures 9 (d), (c), and (b), respectively. Table 6 also presents the numerical values of the test accuracy and test loss for each of these experiments.
> 2. No, the method also works with a large dataset. In Figure 9, all the settings share in common the need for an early stopping signal, i.e., the best performance is achieved at some epoch during training and not at the very last epoch. To be more precise: in settings (a), (b), and (d) of Figure 9, we observe that if the training continues beyond the early stopping signal (the magenta vertical bar given by GD in the 2nd column), the test accuracy decreases and the test loss increases. In setting (c), the test accuracy remains the same beyond that point, whereas the test loss drastically increases, indicating that the classifier is becoming more overconfident in the wrong predictions. Therefore, the use of an early stopping criterion is beneficial in all these settings. We then compare the use of GD to $k$-fold CV as early stopping methods in Table 6 for each of the four settings of Figure 9. We observe that even for networks that are trained on the *entire* dataset (settings (b), (c), and (d)), using GD results in better final test accuracy.
> 3. In our experiments, we do not use data augmentation. Adding these tricks and techniques can indeed improve the final test accuracy. Following the reviewer’s suggestion, and as a sanity check, we have rerun the experiments of Figure 3 with data augmentation, and we have included these new results in the appendix of the revised paper (please refer to Figure 16 of the revised paper).
> Consistent with the rest of the paper, we observe a strong positive correlation ($\rho=0.984$) between the test error and GD for networks that are trained with data augmentation. Moreover, we observe that using data augmentation decreases the values of both gradient disparity and the test error.
> For the data augmentation, we use random crop with padding = 4 and random horizontal flip with probability = 0.5.
> 4. The test error is the classification error computed over the test set. The generalization error (gap) is the difference between the test and train errors. We define these terms in Appendix C. To improve clarity, we added this definition in the caption of Figure 2 as well.
> All the values of the generalization error in Figure 2 (b) are very close to zero (if not exactly zero) and hence generalization error fails to account for the label noise level. This is indeed what we emphasize in this figure: the difference between the train and test errors does not distinguish networks trained on different label noise levels, contrary to gradient disparity. Please see https://ibb.co/kgvs11C  for a zoom-in version of Figure 2 (b).
> 5. We have added the correlation values in Figure 13 (please see the revised paper). Similar figures, e.g. Figures 11, 22, and 24, also include the correlation values. We have also emphasized the correlation to the test loss in Table 9. We observe that even though both Var and GD perform similarly as early stopping criteria, the correlation between GD and the test loss is stronger.

---

### Author Response · Authors · 2020-11-19
**Please see the updated paper**

We thank all the reviewers for their feedback. We have uploaded a revised version of the paper with the changes suggested by the reviewers, which we have addressed in our responses below.

---

### Decision · Program_Chairs · 2021-01-07
**Final Decision**

**Decision:**

Reject

**Comment:**

This paper proposes an early stopping strategy based on the disparity of gradients between two batches from the *training set*. Such a criterion would make the held-out validation set unnecessary. The idea is motivated by theoretical arguments and benchmarked on experiments, but some issues still need to be worked on.

Regarding theory, the theorems here assume implicitly the independence of the gradients computed on two different batches of training data, but the conditions where gradients on independent examples computed *on trained weights* are independent (or close to being independent) should be discussed. Regarding experiments, the protocol is unusual in that the proposed stopping criterion is compared to a stopping criterion relying on k-fold cross-validation, instead of the usual stopping criterion on held-out validation. What is the motivation for this protocol? Why should we expect a small variability in the optimal number of iterations over the k runs?  In addition, the experiments consider a normalized definition of gradient disparity for which no theory is provided. Although there is an interesting correlation between this normalized gradient disparity and generalization error, this link does not seem strong enough to pick the right number of iterations.

Detail:
Still regarding independence, reporting the empirical standard errors on k-fold cross-validation is debatable since it is not related to the theoretical standard error (see e.g. [Bengio et al.: No Unbiased Estimator of the Variance of K-Fold Cross-Validation. J. Mach. Learn. Res. 5: 1089-1105 (2004)])